# A novel ABC fractional-order mathematical model for malaria transmission dynamics incorporating treatment-seeking behavior

Sisay Fikadu Jaleta1*, Gemechis File Duressa2, Chernet Tuge Deressa2

1 Department of Mathematics, Debre Berhan University, Debre Berhan, Ethiopia, 2 Department of Mathematics, Jimma University, Jimma, Ethiopia

* sisayfikadu28@gmail.com, sisayfikadu@dbu.edu.et

## Abstract

Malaria remains a significant global health challenge, particularly in developing countries. This study introduces a novel ABC fractional-order model to analyze malaria transmission dynamics, incorporating treatment-seeking behavior, which includes both treatment at professional health facilities and interventions through indigenous traditional medicine. We conducted a comprehensive analysis of the model, examining the existence and uniqueness of solutions and performing numerical simulations using various mathematical techniques. Our findings reveal that fractional-order effects significantly influence malaria transmission dynamics; specifically, higher fractional orders result in slower increases in susceptible and exposed human populations while leading to more rapid changes in the dynamics of infected populations. Furthermore, the model demonstrates that increasing the rate of treatment at health facilities can substantially reduce the infected population and decrease the reproduction number, thereby facilitating the elimination of the disease within a shorter time frame. Additionally, the study highlights that reliance on traditional medicine without clinical validation may lead to temporary recovery but not complete elimination of the malaria parasite, increasing the risk of relapse and further disease spread. The finding suggests that public health initiatives should encourage collaboration between traditional medicine practitioners and professional healthcare providers to reduce non-standardized treatment risks and improve the effectiveness of malaria management strategies. These contribute to the broader field of epidemiological modeling, offering a robust framework for understanding and mitigating malaria transmission in resource-limited settings.

## 1. Introduction

Malaria is a life-threatening disease caused by Plasmodium parasites, which are transmitted to humans through the bites of infected female Anopheles mosquitoes [1].

**Data availability statement:** All relevant data are within the manuscript and its Supporting information files.

**Funding:** The author(s) received no specific funding for this work.

**Competing interests:** The authors have declared that no competing interests exist.

Despite ongoing efforts to curb its prevalence through vector control, drug treatment, and educational initiatives, it remains a persistent challenge due to its complex transmission dynamics and the sociocultural factors influencing treatment-seeking behaviors. It affects nearly half of the world's population, with 249 million people contracting the disease in 85 countries in 2022, resulting in around 608,000 deaths [2]. Malaria is more common in infants and children under five, pregnant women, and HIV/AIDS patients [3]. The complexity of malaria transmission dynamics, influenced by various factors, necessitates innovative approaches to modeling and analysis [4]. Traditional mathematical models, often based on integer-order derivatives, have provided foundational insights into disease spread and control [5]. However, these models sometimes lack the flexibility to capture the memory and hereditary effects inherent in biological systems and human behavior, which are crucial in understanding and controlling malaria dynamics.

Recent advances in fractional calculus have opened new avenues for modeling complex systems and real-world phenomena more effectively than classical calculus [6–9]. Fractional derivative models are mostly used because time-fractional operators enable memory effects, meaning that a system's reaction is dependent on its past, and space-fractional operators allow for nonlocal and scale effects [10,11]. The memory dynamics and genetic properties present in both biological and engineering systems are captured by fractional-order operators, which also expand the domain of stability [12,13]. In biological modeling, fractional-order terms often arise in the context of systems that exhibit memory or non-integer dynamic behaviors [11]. The biological significance of fractional-order terms can be understood in several ways, as they reflect processes where the rate of change of a system is not simply linear or exponential but involves complex dependencies on past states or behaviors. In epidemiology, fractional-order models are gaining attention because they can better capture certain real-world complexities in disease spread [14]. Unlike traditional models, which assume a relatively simple progression and recovery process, fractional-order models can describe memory effects, where the current state depends not just on recent conditions but also on past states in a weighted way [15]. This adaptability is particularly relevant for malaria, where factors such as delayed immune responses, intermittent treatment adherence, and environmental influences can be better modeled with fractional derivatives. It can also offer a better fit for real data for the various disease models [14,16].

The Atangana-Baleanu-Caputo (ABC) fractional derivative is a relatively recent mathematical operator that provides a novel approach to modeling complex systems [9]. Its main advantage lies in its non-singular and non-local kernel, which makes it particularly suited for modeling processes that involve memory and hereditary properties, common in various real-world phenomena [7,9] The ABC operator uses a Mittag-Leffler function kernel, which helps in capturing memory effects realistically unlike Caputo or Riemann-Liouville derivatives that rely on power-law kernels [17–20] Moreover, this operator is preferred due to its ability to avoid singularities in the kernel while retaining memory effects, resulting in more realistic modeling of memory and hereditary properties in various systems, reducing divergence or overfitting issues

in numerical computation, and enabling accurate descriptions of anomalous diffusion and complex dynamics [7,9,14,21]. The ABC derivative is increasingly used in modeling the spread of diseases such as COVID-19, influenza, and malaria [11,22–24], where it allows researchers to simulate realistic disease progression and recovery dynamics. Furthermore, the ABC derivative is used in various real-life scenarios, including viscoelastic materials, diffusion processes, control systems, epidemiology, finance, and thermodynamics [25–30]. It helps model stress-strain relationships in viscous and elastic materials, describe anomalous diffusion in porous media or biological tissues, design controllers for memory-based systems, capture delayed or non-instantaneous interactions in diseases, represent memory and volatility clustering in financial time series, and describe heat conduction in materials with memory effects. By employing the ABC operator, researchers can more accurately capture the nuances of systems that depend on past states, which is often a limitation in classical models. This makes the ABC operator a powerful tool for improving predictive accuracy and understanding complex systems in real-world applications [11,14,24].

Treatment-seeking behavior, encompassing both professional healthcare services and indigenous traditional medicine, can significantly affect disease outcomes [31]. Traditional medicine, a combination of herbal knowledge, spiritual practices, and rituals, is a significant approach to malaria treatment, especially in resource-limited settings [32–34]. This approach is accessible, affordable, and deeply rooted in cultural practices, making it suitable for rural and impoverished populations. While professional healthcare facilities offer standardized treatments with proven efficacy, traditional medicine remains a primary or supplementary option for many individuals in malaria-endemic regions [35]. However, it also poses risks like misuse and toxicity, lack of safety, dosage unpredictability, and variations in preparation methods due to the non-constant biochemical compositions of the plants used [36]. The absence of a standardized prescription and dosage system in traditional medicine poses a significant risk to anti-malarial drug resistance, posing a significant challenge to malaria control [36]. The authors in [37] explore the role of traditional healers in managing severe malaria among children under five. The study reveals that traditional healers influence care-seeking behaviors, delay access to formal healthcare and relapse, and complicate timely intervention. Additionally, the study in [38] presents the use of traditional herbal medicines for malaria treatment, highlighting their long history and importance in tropical regions. They highlight the effectiveness of remedies like Cryptolepis sanguinolenta and Artemisia annua in reducing parasitemia and resolving symptoms like fever. However, relapses are common, as Artemisia annua achieved good parasite clearance at day 7, and by day 28, only 37% of patients remained parasite-free compared to 86% treated with Quinine (a modern antimalarial drug). This duality in treatment-seeking behavior introduces complexities that are inadequately addressed in existing malaria transmission models.

Employing integer-order derivatives, numerous mathematical models have been devised to explore the intricate dynamics of malaria spread, incorporating factors like latent infection periods, heterogeneity in human-mosquito interactions, immunity levels, susceptibility to malaria, and host-level vulnerability [39–41]. However, traditional mathematical model approaches often fail to capture the delayed or memory-driven processes inherent in treatment-seeking behaviors, relapse dynamics, and the effectiveness of interventions [11,24].

Recently, studies have highlighted the advantages of fractional-order models in epidemiology. For instance, fractional-order derivatives have been used to model the dynamics of diseases such as COVID-19, HIV, and tuberculosis [14,23,42], demonstrating improved accuracy in capturing long-term trends and oscillatory behavior. The studies in [11,24] explored a mathematical model of malaria transmission dynamics using Atangana-Baleanu derivatives, emphasizing the importance of considering memory effects in disease modeling and its Mittag-Leffler kernel, which provides a more accurate representation of real-world processes. Despite these developments, existing models rarely account for the dual pathways of treatment-seeking behavior: professional healthcare services and traditional medicine. This is critical, especially in resource-limited settings where reliance on traditional medicine is widespread.

The novelty of this paper lies in developing a new fractional-order model using the Atangana-Baleanu-Caputo (ABC) derivative to investigate malaria transmission dynamics, explicitly incorporating treatment-seeking behavior. The study's

primary contributions lie in its innovative application of the ABC fractional-order model to malaria, the integration of treatment-seeking behavior into the modeling framework, and suggesting policy recommendations for collaboration between traditional medicine practitioners and professional healthcare providers to enhance malaria control strategies. These contribute to the broader field of epidemiological modeling, offering a robust framework for understanding and mitigating malaria transmission in resource-limited settings.

## 2. Preliminaries

In this section, let us recall the basic definitions of ABC fractional operators and well-known theorems from fractional calculus.

**Definition 1** [43] The gamma function of $x > 0$ is defined by the integral.

$$\Gamma(x) = \int_0^\infty e^{-t} t^{x-1} dt \tag{1}$$

**Definition 2** [16,18,43] The Mittag-Leffler function denoted by $E_\alpha(z)$ and defined as:

$$E_\alpha(z) = \sum_{k=0}^\infty \frac{z^k}{\Gamma(\alpha k + 1)}, \ \alpha, \ z \in \mathbb{C}, \ Re(\alpha) > 0 \tag{2}$$

**Definition 3** [9,14] Let $g : [a, \ b] \to \mathbb{R}$ be a bounded and continuous function. The Atangana-Baleanu fractional derivative in Caputo sense of order $0 < \alpha \leq 1$ is defined as

$${}^{ABC}_{a}D_t^\alpha g(t) = \frac{M(\alpha)}{(1-\alpha)} \int_a^t E_\alpha \left( -\frac{\alpha}{(1-\alpha)} (t-h)^\alpha \right) g'(h) dh \tag{3}$$

Where $M(\alpha)$ is positive and a normalization function given by $M(\alpha) = 1 - \alpha + \frac{\alpha}{\Gamma(\alpha)}$, characterized by $M(0) = M(1) = 1$.

**Definition 4** [11,14] Let $g : [a, \ b] \to \mathbb{R}$ be a bounded and continuous function. The corresponding fractional integral concerning to Atangana-Baleanu fractional order derivative of order $0 < \alpha \leq 1$ is defined as:

$${}^{ABC}_{a}I_t^\alpha g(t) = \frac{(1-\alpha)}{M(\alpha)} g(t) + \frac{\alpha}{M(\alpha)\Gamma(\alpha)} \int_a^t (t-h)^{\alpha-1} g(h) dh \tag{4}$$

**Theorem 1.** Let $g : [a, \ b] \to \mathbb{R}$ be a bounded and continuous function, $a < b$, $\alpha \in [0, \ 1]$, then the following equality on $[a, \ b]$ is satisfied [18].

$${}^{AB}_{a}I_t^\alpha \left( {}^{ABC}_{a}D_t^\alpha g(t) \right) = g(t) - g(a) \tag{5}$$

**Theorem 2.** Let $g : [a, \ b] \to \mathbb{R}$ be a bounded and continuous function. Then, the following results hold as in [9], $\left\| {}^{ABC}_{a}D_t^\alpha g(t) \right\| \leq \frac{M(\alpha)}{(1-\alpha)} \|g(t)\|$, where $\|g(t)\| = \max_{a \leq t \leq b} |g(t)|$.

Moreover, for two functions $g_1, g_2 \in L(a, \ b), b > a$; then the AB derivative satisfies the Lipschitz condition [9]:

$$\left\| {}^{ABC}_{a}D_t^\alpha g_1(t) \leq {}^{ABC}_{a}D_t^\alpha g_2(t) \right\| \leq L \|g_1(t) - g_2(t)\| \tag{6}$$

Where, $0 < \alpha \leq 1$ is the order of fractional derivative.

**Theorem 3.** In the Caputo sense, the Laplace transform of the Atangana-Baleanu fractional derivative is given as [9]:

$$\mathcal{L}\left\{{}^{ABC}_{a}D^{\alpha}_{t}g(t)\right\}(s) = \frac{M(\alpha)}{(1-\alpha)}\frac{s^{\alpha}\mathcal{L}\{g(t)\}(s) - s^{\alpha-1}g(0)}{s^{\alpha} + \frac{\alpha}{1-\alpha}} = \frac{M(\alpha)}{(1-\alpha)}\frac{s^{\alpha}\mathcal{L}\{g(t)\}(s)}{s^{\alpha} + \frac{\alpha}{1-\alpha}} - \frac{M(\alpha)}{(1-\alpha)}\frac{s^{\alpha-1}g(0)}{s^{\alpha} + \frac{\alpha}{1-\alpha}}$$

$$= \frac{M(\alpha)\left(s^{\alpha}G(s) - s^{\alpha-1}g(0)\right)}{s^{\alpha}(1-\alpha) + \alpha}, \quad s > 0 \tag{7}$$

## 3. Formulation and description of the model

The total human population at a time $t$, denoted by $N_h(t)$ is divided into six epidemiological categories in the presence of the disease: Susceptible $S_h$, Exposed $E_h$, Infectious $I_h$, Treatment at health facilities $T_h$, Treatment with indigenous traditional medicines (drug that is prepared by traditional practitioners but is clinically not valid $T_m$) and individuals who recover due to traditional remedies intervention/natural immunity $R_h$. So, the total human population at any time $t \geq 0$, is given by:

$$N_h(t) = S_h(t) + E_h + I_h(t) + T_h(t) + T_m(t) + R_h(t)$$

In a similar manner, the female mosquito populations are divided into three compartments: Susceptible $S_m$, Exposed $E_m$, and Infectious $I_m$. Thus, at any time $t \geq 0$, the total female mosquito populations denoted $N_m(t)$, is given by:

$$N_m(t) = S_m(t) + E_m(t) + I_m(t)$$

Humans enter the susceptible class via natural birth rate $\Lambda_h$, due to the loss of natural immunity from the human treatment class at health facilities with rate $\rho$, by loss of immunity from the recovered class (at a constant rate $\rho_1$), and can be decreased by natural death at a constant rate $\mu$ or infected after a bite from an infectious mosquito and the sporozoites are passed on to them. The transmission rate of infections in a susceptible human population (from an infectious mosquito to a susceptible human) is assumed to be given by a rate, $\lambda_h$. An exposed individual becomes infectious at a constant rate $\theta_h$ and an infectious human $I_h$ who seeking treatment at professional health facility will move to the treatment class $T_h$ with rate $\tau_1$. It is assumed that an individuals who undergoing treatment at health facilities will recover successfully and then return to the susceptible class $S_h$ by loss of immunity (at a constant rate $\rho$). In addition, an infectious human who seeking treatment with traditional medicine without a prescription from health professionals will move to $T_m$ at a constant rate $\tau_2$. We assumed that an individual in a $T_m$ class can recover temporarily with a constant rate $\gamma$, due to natural immunity and traditional medicine interventions; while the remaining of individuals return to a health facilities due to the failure of traditional medicine interventions at a constant progression rate $\omega$. A temporarily recovered human class can be entered into susceptible human class by losing of immunity with constant rate, $\rho_1$ if the merozoites of parasites clear from the blood completely, and back to the infectious class $I_h$ with constant rate $\gamma_1$, otherwise. All human population classes decreased through natural death rate $\mu$ and disease-induced death rates for $I_h$ and $T_m$, $\delta$ and $\psi$, respectively.

Similarly, new mosquitoes are recruited at a constant rate $\Lambda_m$ into the susceptible mosquitoes, $S_m$. When a susceptible mosquito bites an infectious human $I_h$ or human in treatment with traditional medicines, $T_m$, the parasite enters the mosquito, and then moves to the exposed mosquitoes class, $E_m$ by the force of infection, $\lambda_m$. An exposed mosquito will become infectious at a constant rate, $\theta_m$. All mosquito population classes decreased by natural death rate, $\eta$.

We assumed community treatment facilities divide malaria-infected individuals into two groups: those seeking treatment in health facilities and those using traditional medicines without a prescription. Traditional medicine (TM) refers to antimalarial drugs prepared by traditional practitioners, but it lacks clinical validity, quality control, safety measures, standardized dosages, and potential drug interactions. While individuals undergoing traditional treatment may experience temporary recovery due to natural immunity, it is assumed that the merozoites of parasites are not completely eliminated from the bloodstream due to the ineffectiveness of these interventions. Fig 1 shows the dynamics of malaria in both human and mosquito populations, based on the given descriptions and assumptions.

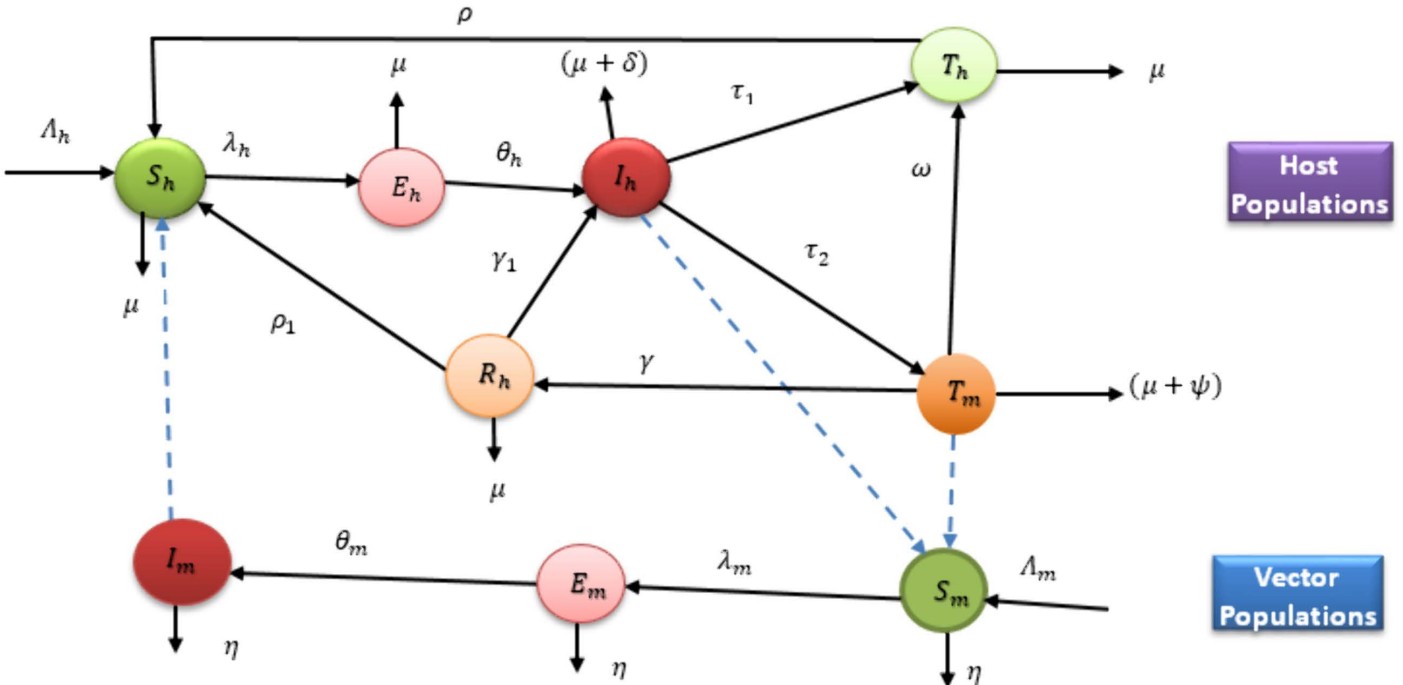

**Fig 1. Flow diagram of the malaria transmission dynamics.** The dashed arrows indicate the direction of the infection and the solid arrows represent the transition from one class to another.

Based on the flow diagram and given assumptions, the mathematical model with integer order used in this study is expressed by the nonlinear ordinary differential equations:

$$\frac{dS_h}{dt}(t) = \Lambda_h - \lambda_h S_h + \rho T_h + \rho_1 R_h - \mu S_h$$

$$\frac{dE_h}{dt}(t) = \lambda_h S_h - (\theta_h + \mu) E_h$$

$$\frac{dI_h}{dt}(t) = \theta_h E_h + \gamma_1 R_h - (\delta + \tau_1 + \tau_2 + \mu) I_h$$

$$\frac{dT_h}{dt}(t) = \tau_1 I_h + \omega T_m - (\rho + \mu) T_h$$

$$\frac{dT_m}{dt}(t) = \tau_2 I_h - (\omega + \gamma + \mu + \psi) T_m$$

$$\frac{dR_h}{dt}(t) = \gamma T_m - (\gamma_1 + \rho_1 + \mu) R_h$$

$$\frac{dS_m}{dt}(t) = \Lambda_m - (\lambda_m + \eta) S_m$$

$$\frac{dE_m}{dt}(t) = \lambda_m S_m - (\theta_m + \eta) E_m$$

$$\frac{dI_m}{dt}(t) = \theta_m E_m - \eta I_m$$

(8)

With an initial conditions $(S_h(0),\ E_h(0),\ I_h(0),\ T_h(0),\ T_m(0),\ R_h(0),\ S_m(0),\ E_m(0),\ I_m(0)) \in \mathbb{R}_+^9$ and where $\lambda_h$ and $\lambda_m$ are the forces of infections from mosquito to human and human to mosquito, respectively given by:

$$\lambda_h\,(I_m) = q\beta_{mh}\frac{I_m}{N_h},\ \ \lambda_m\,(I_h + T_m) = q\alpha_{1m}\frac{I_h}{N_h} + q\alpha_{2m}\frac{T_m}{N_h} \tag{9}$$

The AB fractional order in Caputo sense system of DEs (8) is described as follows:

$$\begin{cases} {}^{ABC}_{\ \ 0}D_t^\alpha S_h(t) = P_1\,(t, S_h(t)) \\ {}^{ABC}_{\ \ 0}D_t^\alpha E_h(t) = P_2\,(t, E_h(t)) \\ {}^{ABC}_{\ \ 0}D_t^\alpha I_h(t) = P_3\,(t, I_h(t)) \\ {}^{ABC}_{\ \ 0}D_t^\alpha T_h(t) = P_4\,(t, T_h(t)) \\ {}^{ABC}_{\ \ 0}D_t^\alpha T_m(t) = P_5\,(t, T_m(t)) \\ {}^{ABC}_{\ \ 0}D_t^\alpha R_h(t) = P_6\,(t, R_h(t)) \\ {}^{ABC}_{\ \ 0}D_t^\alpha S_m(t) = P_7\,(t, S_m(t)) \\ {}^{ABC}_{\ \ 0}D_t^\alpha E_m(t) = P_8\,(t, E_m(t)) \\ {}^{ABC}_{\ \ 0}D_t^\alpha I_m(t) = P_9\,(t, I_m(t)) \end{cases} \tag{10}$$

In which the kernels are provided by:

$$\begin{cases} P_1\,(t, S_h(t)) = \Lambda_h{}^\alpha - (\lambda_h + \mu)S_h + \rho^\alpha T_h + \rho_1{}^\alpha R_h \\ P_2\,(t, E_h(t)) = \lambda_h S_h - (\theta_h{}^\alpha + \mu^\alpha)\,E_h \\ P_3\,(t, I_h(t)) = \theta_h{}^\alpha E_h + \gamma_1{}^\alpha R_h - (\delta^\alpha + \tau_1{}^\alpha + \tau_2{}^\alpha + \mu^\alpha)\,I_h \\ P_4\,(t, T_h(t)) = \tau_1{}^\alpha I_h + \omega^\alpha T_m - (\rho^\alpha + \mu^\alpha)\,T_h \\ P_5\,(t, T_m(t)) = \tau_2{}^\alpha I_h - (\omega^\alpha + \gamma^\alpha + \mu^\alpha + \psi^\alpha)\,T_m \\ P_6\,(t, R_h(t)) = \gamma^\alpha T_m - (\gamma_1{}^\alpha + \rho_1{}^\alpha + \mu^\alpha)\,R_h \\ P_7\,(t, S_m(t)) = \Lambda_m{}^\alpha - (\lambda_m + \eta^\alpha)\,S_m \\ P_8\,(t, E_m(t)) = \lambda_m S_m - (\theta_m{}^\alpha + \eta^\alpha)\,E_m \\ P_9\,(t, I_m(t)) = \theta_m{}^\alpha E_m - \eta^\alpha I_m \end{cases} \tag{11}$$

With an initial conditions $S_h(0) \geq 0,\ E_h(0) \geq 0,\ I_h(0) \geq 0,\ T_h(0) \geq 0,\ T_m(0) \geq 0,\ R_h(0) \geq 0,\ S_m(0) \geq 0,\ E_m(0) \geq 0,\ I_m(0) \geq 0$ and where ${}^{ABC}_{\ \ 0}D_t^\alpha$ is $ABC$ fractional derivative of order $\alpha$, $\lambda_h$ and $\lambda_m$ are the forces of infections from mosquito to human and human to mosquito, respectively given by:

**Table 1. The state variable descriptions of the model.**

| Variables | Descriptions |
|---|---|
| $S_h(t)$ | Susceptible humans |
| $E_h(t)$ | Exposed humans |
| $I_h(t)$ | Infected humans |
| $T_h(t)$ | Treatment at a health facility (infectious individuals who seek treatment at health facilities) |
| $T_m(t)$ | Treatment with indigenous traditional medicines((infectious individuals who seek treatment with TM) |
| $R_h(t)$ | Recovered humans |
| $S_m(t)$ | Susceptible mosquitoes |
| $E_m(t)$ | Exposed mosquitoes |
| $I_m(t)$ | Infected mosquitoes |

**Table 2. Parameters description of the model with their values.**

| Parameters | Descriptions | Units | Values | References |
|---|---|---|---|---|
| $\Lambda_h$ | Recruitment rate for humans | /month | 0.000215 | [39] |
| $\Lambda_m$ | Recruitment rate for mosquitoes | /month | 0.07 | [39] |
| $\mu$ | Natural death rate of humans | /month | 0.000045 | [40] |
| $\eta$ | Natural death rate of mosquitoes | /month | 0.0477 | [40] |
| $q$ | Average per capita biting rate of mosquitoes | /month | 0.33 | [40] |
| $\alpha_{1m}$ | Probability of tranission of infection from $I_h$ to $S_m$ | – | 0.8333 | [41] |
| $\alpha_{2m}$ | Probability of transmission of infection from $T_m$ to $S_m$ | – | 0.0833 | [41] |
| $\beta_{mh}$ | Probability of transmission of infection from $I_m$ to $S_h$ | – | 0.02 | [41] |
| $\theta_h$ | Latent period in humans | /month | 0.1 | [40] |
| $\theta_m$ | Latent period in mosquitoes | /month | 0.08 | [40] |
| $\delta$ | Disease induced death rate of $I_h$ class | /month | 0.0018 | [40] |
| $\psi$ | Disease induced death rate of $I_h$ class | /month. | 0.0001 | Assumed |
| $\tau_1$ | From infectious human class to humans treatment class at health facilities, constant treatment rate of humans | /month | [0, 1] | Estimated from [44,45] |
| $\tau_2$ | From infectious human class to humans treatment class with traditional medicines | /month | 0.605 | Assumed |
| $\rho$ | From humantreatment class at health facilities to a susceptible human class, rate of loss of immunity | /month | 0.0166 | [44,45] |
| $\gamma$ | Recovery rate due to natural immunity and the use of traditional medicines | /month | 0.0065 | Assumed |
| $\omega$ | From humans treatment with traditional medicines class to treatment at health facilities, progression rate due to ineffectiveness of traditional medicines | /month | 0.01 | Assumed |
| $\gamma_1.$ | From a recovered human class to an infectious human class, relapse rate | /month | 0.1 | [44] |
| $\rho_1$ | From a recovered human class to a susceptible human class, rate of loss of immunity | /month | 0.0146 | [41] |

$$\lambda_h(I_m) = q^\alpha \beta_{mh}{}^\alpha \frac{I_m}{N_h}, \quad \lambda_m(I_h + T_m) = q^\alpha \alpha_{1m}{}^\alpha \frac{I_h}{N_h} + q^\alpha \alpha_{2m}{}^\alpha \frac{T_m}{N_h} \tag{12}$$

Equation (10) can be rewritten as:

$$^{ABC}_0 D_t^\alpha X(t) = P_i(t, X(t)), \quad X(0) = X_0$$

Where, $X(t) = (S_h(t), E_h(t), I_h(t), T_h(t), T_m(t), R_h(t), S_m(t), E_m(t), I_m(t))$

## 4. Model analysis

The model represented by the Equation (10) will be analyzed in the feasible region and since the model represents the populations all the state variables and the parameters are assumed positive.

**Theorem 4**. For all time $t \geq 0$, the solutions of system (10) with $S_h(0) > 0$, $E_h(0) > 0$, $I_h(0) > 0$, $T_h(0) > 0$, $T_s(0) > 0$, $R_h(0) > 0$, $S_m(0) > 0$, $E_m(0) > 0$, $I_m(0) > 0$ are all positive.

Consider the following lemma in order to demonstrate positivity.

**Lemma 1** (Generalized Mean Value Theorem, [31]). Let $h(x) \in C[a, b]$ and $^{ABC}_0 D_t^\alpha h(x) \in C[a, b]$ for $\alpha \in (0, 1]$. Then $h(x) = h(a) + \frac{1}{\Gamma(\alpha)}{}^{ABC}_0 D_t^\alpha h(\zeta)(x-a)^\alpha$, where $\zeta \in [0, x]$, $\forall x \in (a, b]$.

Recall that by Lemma 1, if $h(x) \in [0, b]$, $^{ABC}_0 D_t^\alpha h(x) \in [0, b]$, $^{ABC}_0 D_t^\alpha h(x) \geq 0$, $\forall x \in (0, b]$, when $\alpha \in (0, 1]$, $h(x)$ is constant or increasing function, and if $^{ABC}_0 D_t^\alpha h(x) \leq 0$, $\forall x \in (0, b]$, then the function $h(x)$ is constant or decreasing, $\forall x \in (0, b]$.

Let us demonstrate that $\Omega$ is positively invariant. Using the lemma 1, we obtain

$$\begin{cases} {}^{ABC}_0D^\alpha_t S_h|_{S_h=0} = \Lambda_h + \rho T_h + \rho_1 R_h > 0, \\ {}^{ABC}_0D^\alpha_t E_h|_{E_h=0} = \lambda_h S_h \geq 0, \\ {}^{ABC}_0D^\alpha_t I_h|_{I_h=0} = \theta_h E_h + \gamma_1 R_h \geq 0, \\ {}^{ABC}_0D^\alpha_t T_h|_{T_h=0} = \tau_1 I_h + \omega T_m \geq 0, \\ {}^{ABC}_0D^\alpha_t T_m|_{T_m=0} = \tau_2 I_h \geq 0, \\ {}^{ABC}_0D^\alpha_t R_h|_{R_h=0} = \gamma T_m \geq 0, \\ {}^{ABC}_0D^\alpha_t S_m|_{S_m=0} = \Lambda_m > 0, \\ {}^{ABC}_0D^\alpha_t E_m|_{E_m=0} = \lambda_m S_m \geq 0, \\ {}^{ABC}_0D^\alpha_t I_m|_{I_m=0} = \theta_m E_m \geq 0. \end{cases} \tag{13}$$

Thus, (13) indicates that the feasible region in $\Omega$ is positively invariant for model (10), as inferred from lemma 1, ensuring the solution remains in $\mathbb{R}^6_+ \times \mathbb{R}^3_+$.

**Theorem 5**. The feasible region $\Omega$ defined by:

$$\Omega_h = \left\{ (S_h,\ E_h,\ I_h,\ T_h, T_m,\ R_h) \in \mathbb{R}^6_+ : (S_h + E_h + I_h + T_h + T_m + R_h) \leq \frac{\Lambda_h}{\mu} \right\}$$

$$\Omega_m = \left\{ (S_m,\ E_m,\ I_m) \in \mathbb{R}^3_+ : (S_m + E_m + I_m) \leq \Lambda_m/\eta \right\},$$

With initial condition $S_h(0) > 0,\ E_h(0) > 0, I_h(0) > 0,\ T_h(0) > 0,\ T_s(0) > 0,\ R_h(0) > 0,\ S_m(0) > 0,\ E_m(0) > 0, I_m(0) > 0$ is positively invariant for the system (8) in $\mathbb{R}^9_+$.

**Proof:** The boundedness of the fractional model solutions is established by summing the first six equations of the model (10), resulting in the total human population:

$${}^{ABC}_0D^\alpha_t N_h(t) = (\Lambda_h - \mu S_h - \mu E_h - (\delta + \mu)I_h - \mu T_h - (\mu + \psi)T_m - \mu R_h) \leq \Lambda_h - \mu N_h(t)$$

$$\text{}^{ABC}_0D^\alpha_t N_h(t) + \mu N_h(t) \leq \Lambda_h \tag{14}$$

In the same manner, the total mosquito population can be determined by adding the last three equations in the system (10) given as:

$${}^{ABC}_0D^\alpha_t N_m(t) + \eta N_m(t) \leq \Lambda_m \tag{15}$$

Applying the Laplace transform to equation (14), we obtain

$$\mathcal{L}\left({}^{ABC}_0D^\alpha_t N_h(t) + \mu N_h(t)\right) \leq \mathcal{L}\left(\Lambda_h\right),$$

$$\mathcal{L}(N_h)\left((1-\varepsilon)s^\alpha - \frac{\varepsilon\alpha}{1-\alpha}\right) - s^{\alpha-1}N_h(0) \leq \frac{1-\alpha}{M(\alpha)}\left(s^\alpha + \frac{\alpha}{1-\alpha}\right)\frac{\Lambda_h}{s},$$

$$\mathcal{L}\left(N_h\right) \leq \left(1 - \frac{\varepsilon\alpha}{(1-\varepsilon)(1-\alpha)}s^{-\alpha}\right)^{-}\left[\frac{1-\alpha}{(1-\varepsilon)\bar{M}(\alpha)}\left(1 + \frac{\alpha}{1-\alpha}s^{-\alpha}\right)\frac{\Lambda_h}{s} + N_h(0)\frac{1}{(1-\varepsilon)s}\right],$$

Where $\varepsilon = \frac{-\mu(1-\alpha)}{M(\alpha)}$.

The solution, obtained by applying the inverse Laplace transform and following the work [46], is as follows:

$$N_h(t) = \frac{\Lambda_h}{\mu} - \frac{\Lambda_h}{\mu(1-\varepsilon)}\frac{d}{dt}\int_0^t E_\alpha\left(\frac{\varepsilon\alpha}{(1-\varepsilon)(1-\alpha)}(t-x)^\alpha dx\right) + \frac{1}{(1-\varepsilon)}E_\alpha\left(\frac{\varepsilon\alpha}{(1-\varepsilon)(1-\alpha)}t^\alpha\right)N_h(0),$$

Where $E_{\alpha,\,\beta}$ refers to the Mittag-Leffler function and it exhibits asymptotic behavior.

$$E_{\alpha,\,\beta}(Z) \approx \sum_{k=1}^{\omega}\frac{z^{-k}}{\Gamma(\beta-\alpha k)} + O\left(|z|^{-1-\omega}\right), \quad |z| \to \infty, \ \frac{\alpha\pi}{2} < |arg(z)| \leq \pi,$$

It can be observed that $N_h(t) \to \frac{\Lambda_h}{\mu}$ as $t \to \infty$. As a result, it is biologically feasible region and the total human population is bounded. Similarly, the mosquito population, $N_m(t) \to \frac{\Lambda_m}{\eta}$ as $t \to \infty$, is also bounded.

Therefore, $\Omega = \Omega_h \times \Omega_m \subset \mathbb{R}_+^9$ is biologically feasible region of model (10).

## 4.1. Existence and uniqueness of solutions

To show the existence of the solution to model (10), we use the Banach fixed point theorem, with further study on fixed points and contractions referred to in [47] and its references. By applying the Laplace transform to both sides of Equation (10)'s first equation, we obtain,

$$\mathcal{L}\{^{ABC}_{\ 0}D_t^\alpha S_h(t)\}(s) = \mathcal{L}\{P_1\left(t, S_h(t)\right)\}(s), \ s > 0 \tag{16}$$

Moreover, theorem 3 gives us that,

$$\frac{M(\alpha)}{(1-\alpha)}\frac{s^\alpha \mathcal{L}\{S_h(t)\}(s) - s^{\alpha-1}S_h(0)}{s^\alpha + \frac{\alpha}{1-\alpha}} = \mathcal{L}\{g(t)\}(s), \ s > 0$$

Where $g(t) = P_1\left(t, S_h(t)\right)$, which implies that,

$$\mathcal{L}\left\{S_h(t)\right\}(s) = \frac{1}{s}S_h(0) + \frac{(1-\alpha)}{M(\alpha)}\mathcal{L}\left\{g(t)\right\}(s) + \frac{\alpha}{s^\alpha M(\alpha)}\mathcal{L}\left\{g(t)\right\}(s) \tag{17}$$

After using the inverse Laplace transform on both side of Equation (17), we obtain the equation that follows:

$$S_h(t) = S_h(0) + \frac{(1-\alpha)}{M(\alpha)}g(t) + \mathcal{L}^{-1}\left\{\frac{\alpha}{s^\alpha M(\alpha)}[g(t)](s)\right\}(t) \tag{18}$$

The final component in Equation (11) expressed as follows:

$$\mathcal{L}^{-1}\left\{\frac{\alpha}{s^\alpha M(\alpha)}[g(t)](s)\right\}(t) = \mathcal{L}^{-1}\left\{F(s)G(s)\right\}(t), \tag{19}$$

Where $F(s) = \frac{\alpha}{s^\alpha M(\alpha)} = \frac{\alpha}{M(\alpha)}\mathcal{L}\{\frac{t^{\alpha-1}}{s^\alpha}\}$ and $G(s) = \mathcal{L}\{g(t)\}(s)$.

Thus, by using the convolution theorem, we obtain the following equation:

$$\mathcal{L}^{-1}\left\{\frac{\alpha}{s^{\alpha}M(\alpha)}\left[g(t)\right](s)\right\}(t) = \frac{\alpha}{M(\alpha)\Gamma(\alpha)}\int_a^t g(\tau)(t-\tau)^{\alpha-1}d\tau \qquad (20)$$

Therefore, by using (20), Equation (18) becomes

$$S_h(t) - S_h(0) = \frac{(1-\alpha)}{M(\alpha)}P_1\left(t, S_h(t)\right) + \frac{\alpha}{M(\alpha)\Gamma(\alpha)}\int_a^t P_1\left(\tau, S_h\right)(t-\tau)^{\alpha-1}d\tau \qquad (21)$$

In similar manner, we have

$$\begin{cases} E_h(t) - E_h(0) = \frac{(1-\alpha)}{M(\alpha)}P_2\left(t, E_h(t)\right) + \frac{\alpha}{M(\alpha)\Gamma(\alpha)}\int_a^t P_2\left(\tau, E_h\right)(t-\tau)^{\alpha-1}d\tau \\ I_h(t) - I_h(0) = \frac{(1-\alpha)}{M(\alpha)}P_3\left(t, I_h(t)\right) + \frac{\alpha}{M(\alpha)\Gamma(\alpha)}\int_a^t P_3\left(\tau, I_h\right)(t-\tau)^{\alpha-1}d\tau \\ T_h(t) - T_h(0) = \frac{(1-\alpha)}{M(\alpha)}P_4\left(t, T_h(t)\right) + \frac{\alpha}{M(\alpha)\Gamma(\alpha)}\int_a^t P_4\left(\tau, T_h\right)(t-\tau)^{\alpha-1}d\tau \\ T_m(t) - T_m(0) = \frac{(1-\alpha)}{M(\alpha)}P_5\left(t, T_m(t)\right) + \frac{\alpha}{M(\alpha)\Gamma(\alpha)}\int_a^t P_5\left(\tau, T_m\right)(t-\tau)^{\alpha-1}d\tau \\ R_h(t) - R_h(0) = \frac{(1-\alpha)}{M(\alpha)}P_6\left(t, R_h(t)\right) + \frac{\alpha}{M(\alpha)\Gamma(\alpha)}\int_a^t P_6\left(\tau, R_h\right)(t-\tau)^{\alpha-1}d\tau \\ S_m(t) - S_m(0) = \frac{(1-\alpha)}{M(\alpha)}P_7\left(t, S_m(t)\right) + \frac{\alpha}{M(\alpha)\Gamma(\alpha)}\int_a^t P_7\left(\tau, S_m\right)(t-\tau)^{\alpha-1}d\tau \\ E_m(t) - E_m(0) = \frac{(1-\alpha)}{M(\alpha)}P_8\left(t, E_m(t)\right) + \frac{\alpha}{M(\alpha)\Gamma(\alpha)}\int_a^t P_8\left(\tau, E_m\right)(t-\tau)^{\alpha-1}d\tau \\ I_m(t) - I_m(0) = \frac{(1-\alpha)}{M(\alpha)}P_9\left(t, I_m(t)\right) + \frac{\alpha}{M(\alpha)\Gamma(\alpha)}\int_a^t P_9\left(\tau, I_m\right)(t-\tau)^{\alpha-1}d\tau \end{cases}$$

Let consider the set $B = H(J) \times H(J) \times H(J) \times H(J) \times H(J) \times H(J) \times H(J) \times H(J) \times H(J)$, where $H(J) = C[0, \tau]$ is the Banach space of real-valued continuous functions defined on an interval $J = [0, \tau]$ with the corresponding norm defined by:

$$\|(S_h, E_h, I_h, T_h, T_m, R_h, S_m, E_m, I_m)\| = \|S_h\| + \|E_h\| + \|I_h\| + \|T_h\| + \|T_m\| + \|R_h\| + \|S_m\| + \|E_m\| + \|I_m\|$$

Where, $\|S_h\| = \underset{t\in[0, \tau]}{Sup}|S_h(t)| = m_1$, $\|E_h\| = \underset{t\in[0, \tau]}{Sup}|E_h(t)| = m_2$, $\|I_h\| = \underset{t\in[0, \tau]}{Sup}|I_h(t)| = m_3$, $\|T_h\| = \underset{t\in[0, \tau]}{Sup}|T_h(t)| = m_4$, $\|T_m\| = \underset{t\in[0, \tau]}{Sup}|T_m(t)| = m_5$, $\|R_h\| = \underset{t\in[0, \tau]}{Sup}|R_h(t)| = m_6$, $\|S_m\| = \underset{t\in[0, \tau]}{Sup}|S_m(t)| = m_7$, $\|E_m\| = \underset{t\in[0, \tau]}{Sup}|E_m(t)| = m_8$, $\|I_m\| = \underset{t\in[0, \tau]}{Sup}|I_m(t)| = m_9$.

**Theorem 6** [18], Lipschitz condition and contraction). F each of the kernels $P_1$, $P_2$, $P_3$, $P_4$, $P_5$, $P_6$, $P_7$, $P_8$, $P_9$ in (10), there exist $L_i > 0$, $i = 1, 2, 3, \ldots 9$, such that,

$$\|P_1\left(t, S_h\right) - P_1\left(t, S_{h1}\right)\| \le L_1\|S_h(t) - S_{h1}(t)\|, \|P_2\left(t, E_h\right) - P_2\left(t, E_{h1}\right)\| \le L_2\|E_h(t) - E_{h1}(t)\|,$$

$$\|P_3\left(t, I_h\right) - P_3\left(t, I_{h1}\right)\| \le L_3\|I_h(t) - I_{h1}(t)\|, \|P_4\left(t, T_h\right) - P_4\left(t, T_{h1}\right)\| \le L_4\|T_h(t) - T_{h1}(t)\|,$$

$$\|P_5\left(t, T_m\right) - P_5\left(t, T_{m1}\right)\| \le L_5\|T_m(t) - T_{m1}(t)\|, \|P_6\left(t, R_h\right) - P_6\left(t, R_{h1}\right)\| \le L_6\|R_h(t) - R_{h1}(t)\|,$$

$$\|P_7\left(t, S_m\right) - P_7\left(t, S_{m1}\right)\| \le L_7\|S_m(t) - S_{m1}(t)\|, \|P_8\left(t, E_m\right) - P_8\left(t, E_{m1}\right)\| \le L_8\|E_m(t) - E_{m1}(t)\|,$$

$$\|P_9\left(t, I_m\right) - P_1\left(t, I_{m1}\right)\| \le L_9\|I_m(t) - I_{m1}(t)\|,$$

and are contractions for $0 \leq L_i < 1$

**Proof**: $\|P_1(t, S_h) - P_1(t, S_{h1})\| = \|(\Lambda_h - (\lambda_h + \mu)S_h + \rho T_h + \rho_1 R_h) - (\Lambda_h - (\lambda_h + \mu)S_{h1} + \rho T_h + \rho_1 R_h)\|$

$= \|(-(\lambda_h + \mu)S_h) + (\lambda_h + \mu)S_{h1}\|$

$\leq (\lambda_h + \mu)\|S_{h1} - S_h\|$

$\leq L_1\|S_{h1} - S_h\|$, where $L_1 = (\lambda_h + \mu)$

The Lipschitz condition is satisfied by $P_1(t, S_h)$ with Lipschitz constant $L_1 = (\lambda_h + \mu)$. Furthermore, if $0 \leq L_1 < 1$, then $P_1(t, S_h)$ is a contraction.

Similarly, we can show the existence of $L_i$, $i = 2, 3, 4, 5, 6, 7, 8, 9$ and a contraction principle for $P_2(t, E_h)$, $P_3(t, I_h)$, $P_4(t, T_h)$, $P_5(t, T_m)$, $P_6(t, R_h)$, $P_7(t, S_m)$, $P_8(t, E_m)$, $P_9(t, I_m)$, $0 \leq L_i < 1$.

Let consider the following recursive form for any positive integer $n$:

$$X_n(t) = \frac{(1-\alpha)}{M(\alpha)}P_i(t, X_{n-1}(t)) + \frac{\alpha}{M(\alpha)\Gamma(\alpha)}\int_{t_0}^t P_i(\tau, X_{n-1}(\tau))(t-\tau)^{\alpha-1}d\tau \tag{22}$$

Then we represent the difference between the successive terms by using the recursive formula in (22).

Thus, to solve the fractional order model by using numerical methods, we can define the recursive form of (10), where $t = t_n$, $n = 1, 2, 3, \ldots$

$$\begin{cases} S_{hn}(t) = \frac{(1-\alpha)}{M(\alpha)}P_1(t, S_{hn-1}) + \frac{\alpha}{M(\alpha)\Gamma(\alpha)}\int_0^t P_1(\tau, S_{hn-1})(t-\tau)^{\alpha-1}d\tau \\ E_{hn}(t) = \frac{(1-\alpha)}{M(\alpha)}P_2(t, E_{hn-1}) + \frac{\alpha}{M(\alpha)\Gamma(\alpha)}\int_0^t P_2(\tau, E_{hn-1})(t-\tau)^{\alpha-1}d\tau \\ I_{hn}(t) = \frac{(1-\alpha)}{M(\alpha)}P_3(t, I_{hn-1}) + \frac{\alpha}{M(\alpha)\Gamma(\alpha)}\int_0^t P_3(\tau, I_{hn-1})(t-\tau)^{\alpha-1}d\tau \\ T_{hn}(t) = \frac{(1-\alpha)}{M(\alpha)}P_4(t, T_{hn-1}) + \frac{\alpha}{M(\alpha)\Gamma(\alpha)}\int_0^t P_4(\tau, T_{hn-1})(t-\tau)^{\alpha-1}d\tau \\ T_{mn}(t) = \frac{(1-\alpha)}{M(\alpha)}P_5(t, T_{mn-1}) + \frac{\alpha}{M(\alpha)\Gamma(\alpha)}\int_0^t P_5(\tau, T_{mn-1})(t-\tau)^{\alpha-1}d\tau \\ R_{hn}(t) = \frac{(1-\alpha)}{M(\alpha)}P_6(t, R_{hn-1}) + \frac{\alpha}{M(\alpha)\Gamma(\alpha)}\int_0^t P_6(\tau, R_{hn-1})(t-\tau)^{\alpha-1}d\tau \\ S_{mn}(t) = \frac{(1-\alpha)}{M(\alpha)}P_7(t, S_{mn-1}) + \frac{\alpha}{M(\alpha)\Gamma(\alpha)}\int_0^t P_7(\tau, S_{mn-1})(t-\tau)^{\alpha-1}d\tau \\ E_{mn}(t) = \frac{(1-\alpha)}{M(\alpha)}P_8(t, E_{mn-1}) + \frac{\alpha}{M(\alpha)\Gamma(\alpha)}\int_0^t P_8(\tau, E_{mn-1})(t-\tau)^{\alpha-1}d\tau \\ I_{mn}(t) = \frac{(1-\alpha)}{M(\alpha)}P_9(t, I_{mn-1}) + \frac{\alpha}{M(\alpha)\Gamma(\alpha)}\int_0^t P_9(\tau, I_{mn-1})(t-\tau)^{\alpha-1}d\tau \end{cases} \tag{23}$$

With an initial conditions $S_{h0} = S_h(0)$, $E_{h0} = E_h(0)$, $I_{h0} = I_h(0)$, $T_{h0} = T_h(0)$, $T_{m0} = T_m(0)$, $R_{h0} = R_h(0)$, $S_{m0} = S_m(0)$, $E_{m0} = E_m(0)$, $I_{m0} = I_m(0)$.

The differences between successive terms in (23) are expressed as follows:

$$\begin{cases} D_{1n}(t) = S_{hn}(t) - S_{hn-1}(t) = \frac{(1-\alpha)}{M(\alpha)}(P_1(t, S_{hn-1}) - P_1(t, S_{hn-2})) + \frac{\alpha}{M(\alpha)\Gamma(\alpha)}\int_0^t (P_1(\tau, S_{hn-1}) - P_1(\tau, S_{hn-2}))(t-\tau)^{\alpha-1}d\tau \\ D_{2n}(t) = E_{hn}(t) - E_{hn-1}(t) = \frac{(1-\alpha)}{M(\alpha)}(P_2(t, E_{hn-1}) - P_2(t, E_{hn-2})) + \frac{\alpha}{M(\alpha)\Gamma(\alpha)}\int_0^t (P_2(\tau, E_{hn-1}) - P_2(\tau, E_{hn-2}))(t-\tau)^{\alpha-1}d\tau \\ D_{3n}(t) = I_{hn}(t) - I_{hn-1}(t) = \frac{(1-\alpha)}{M(\alpha)}(P_3(t, I_{hn-1}) - P_3(t, I_{hn-2})) + \frac{\alpha}{M(\alpha)\Gamma(\alpha)}\int_0^t (P_3(\tau, I_{hn-1}) - P_3(\tau, I_{hn-2}))(t-\tau)^{\alpha-1}d\tau \\ D_{4n}(t) = T_{hn}(t) - T_{hn-1}(t) = \frac{(1-\alpha)}{M(\alpha)}(P_4(t, T_{hn-1}) - P_4(t, T_{hn-2})) + \frac{\alpha}{M(\alpha)\Gamma(\alpha)}\int_0^t (P_4(\tau, T_{hn-1}) - P_4(\tau, T_{hn-2}))(t-\tau)^{\alpha-1}d\tau \\ D_{5n}(t) = T_{mn}(t) - T_{mn-1}(t) = \frac{(1-\alpha)}{M(\alpha)}(P_5(t, T_{mn-1}) - P_5(t, T_{mn-2})) + \frac{\alpha}{M(\alpha)\Gamma(\alpha)}\int_0^t (P_5(\tau, T_{mn-1}) - P_5(\tau, T_{mn-2}))(t-\tau)^{\alpha-1}d\tau \\ D_{6n}(t) = R_{hn}(t) - R_{hn-1}(t) = \frac{(1-\alpha)}{M(\alpha)}(P_6(t, R_{hn-1}) - P_6(t, R_{hn-2})) + \frac{\alpha}{M(\alpha)\Gamma(\alpha)}\int_0^t (P_6(\tau, R_{hn-1}) - P_6(\tau, R_{hn-2}))(t-\tau)^{\alpha-1}d\tau \\ D_{7n}(t) = S_{mn}(t) - S_{mn-1}(t) = \frac{(1-\alpha)}{M(\alpha)}(P_7(t, S_{mn-1}) - P_7(t, S_{mn-2})) + \frac{\alpha}{M(\alpha)\Gamma(\alpha)}\int_0^t (P_7(\tau, S_{mn-1}) - P_7(\tau, S_{mn-2}))(t-\tau)^{\alpha-1}d\tau \\ D_{8n}(t) = E_{mn}(t) - E_{mn-1}(t) = \frac{(1-\alpha)}{M(\alpha)}(P_8(t, E_{mn-1}) - P_8(t, E_{mn-2})) + \frac{\alpha}{M(\alpha)\Gamma(\alpha)}\int_0^t (P_8(\tau, E_{mn-1}) - P_8(\tau, E_{mn-2}))(t-\tau)^{\alpha-1}d\tau \\ D_{9n}(t) = I_{mn}(t) - I_{mn-1}(t) = \frac{(1-\alpha)}{M(\alpha)}(P_9(t, I_{mn-1}) - P_9(t, I_{mn-2})) + \frac{\alpha}{M(\alpha)\Gamma(\alpha)}\int_0^t (P_9(\tau, I_{mn-1}) - P_9(\tau, I_{mn-2}))(t-\tau)^{\alpha-1}d\tau \end{cases} \tag{24}$$

Applying the norm on both sides of each Equation in (24), we obtain.

$$
\begin{cases}
\|D_{1n}(t)\| = \|S_{hn}(t) - S_{hn-1}(t)\| = \frac{(1-\alpha)}{M(\alpha)} \|P_1(t, S_{hn-1}) - P_1(t, S_{hn-2})\| \\
\qquad + \frac{\alpha}{M(\alpha)\Gamma(\alpha)} \int_0^t \|P_1(\tau, S_{hn-1}) - P_1(\tau, S_{hn-2})\| (t-\tau)^{\alpha-1} d\tau \\
\|D_{2n}(t)\| = \|E_{hn}(t) - E_{hn-1}(t)\| = \frac{(1-\alpha)}{M(\alpha)} \|P_2(t, E_{hn-1}) - P_2(t, E_{hn-2})\| \\
\qquad + \frac{\alpha}{M(\alpha)\Gamma(\alpha)} \int_0^t \|P_2(\tau, E_{hn-1}) - P_2(\tau, E_{hn-2})\| (t-\tau)^{\alpha-1} d\tau \\
\|D_{3n}(t)\| = \|I_{hn}(t) - I_{hn-1}(t)\| = \frac{(1-\alpha)}{M(\alpha)} \|P_3(t, I_{hn-1}) - P_3(t, I_{hn-2})\| \\
\qquad + \frac{\alpha}{M(\alpha)\Gamma(\alpha)} \int_0^t \|P_3(\tau, I_{hn-1}) - P_3(\tau, I_{hn-2})\| (t-\tau)^{\alpha-1} d\tau \\
\|D_{4n}(t)\| = \|T_{hn}(t) - T_{hn-1}(t)\| = \frac{(1-\alpha)}{M(\alpha)} \|P_4(t, T_{hn-1}) - P_4(t, T_{hn-2})\| \\
\qquad + \frac{\alpha}{M(\alpha)\Gamma(\alpha)} \int_0^t \|P_4(\tau, T_{hn-1}) - P_4(\tau, T_{hn-2})\| (t-\tau)^{\alpha-1} d\tau \\
\|D_{5n}(t)\| = \|T_{mn}(t) - T_{mn-1}(t)\| = \frac{(1-\alpha)}{M(\alpha)} \|P_5(t, T_{mn-1}) - P_5(t, T_{mn-2})\| \\
\qquad + \frac{\alpha}{M(\alpha)\Gamma(\alpha)} \int_0^t \|P_5(\tau, T_{mn-1}) - P_5(\tau, T_{mn-2})\| (t-\tau)^{\alpha-1} d\tau \\
\|D_{6n}(t)\| = \|R_{hn}(t) - R_{hn-1}(t)\| = \frac{(1-\alpha)}{M(\alpha)} \|P_6(t, R_{hn-1}) - P_6(t, R_{hn-2})\| \\
\qquad + \frac{\alpha}{M(\alpha)\Gamma(\alpha)} \int_0^t \|P_6(\tau, R_{hn-1}) - P_6(\tau, R_{hn-2})\| (t-\tau)^{\alpha-1} d\tau \\
\|D_{7n}(t)\| = \|S_{mn}(t) - S_{mn-1}(t)\| = \frac{(1-\alpha)}{M(\alpha)} \|P_7(t, S_{mn-1}) - P_7(t, S_{mn-2})\| \\
\qquad + \frac{\alpha}{M(\alpha)\Gamma(\alpha)} \int_0^t \|P_7(\tau, S_{mn-1}) - P_7(\tau, S_{mn-2})\| (t-\tau)^{\alpha-1} d\tau \\
\|D_{8n}(t)\| = \|E_{mn}(t) - E_{mn-1}(t)\| = \frac{(1-\alpha)}{M(\alpha)} \|P_8(t, E_{mn-1}) - P_8(t, E_{mn-2})\| \\
\qquad + \frac{\alpha}{M(\alpha)\Gamma(\alpha)} \int_0^t \|P_8(\tau, E_{mn-1}) - P_8(\tau, E_{mn-2})\| (t-\tau)^{\alpha-1} d\tau \\
\|D_{9n}(t)\| = \|I_{mn}(t) - I_{mn-1}(t)\| = \frac{(1-\alpha)}{M(\alpha)} \|P_9(t, I_{mn-1}) - P_9(t, I_{mn-2})\| \\
\qquad + \frac{\alpha}{M(\alpha)\Gamma(\alpha)} \int_0^t \|P_9(\tau, I_{mn-1}) - P_9(\tau, I_{mn-2})\| (t-\tau)^{\alpha-1} d\tau
\end{cases}
\tag{25}
$$

Moreover, the first equality in (25) can be simplified to the following expressions:

$$
\|D_{1n}(t)\| = \|S_{hn}(t) - S_{hn-1}(t)\| \le \frac{(1-\alpha)}{M(\alpha)} \|P_1(t, S_{hn-1}) - P_1(t, S_{hn-2})\|
$$

$$
+ \frac{\alpha}{M(\alpha)\Gamma(\alpha)} \int_0^t \|P_1(\tau, S_{hn-1}) - P_1(\tau, S_{hn-2})\| (t-\tau)^{\alpha-1} d\tau
$$

$$
\le \frac{(1-\alpha)}{M(\alpha)} L_1 \|S_{hn-1} - S_{hn-2}\| + \frac{\alpha}{M(\alpha)\Gamma(\alpha)} L_1 \int_{t_0}^t \|S_{hn-1} - S_{hn-2}\| (t-\tau)^{\alpha-1} d\tau
$$

$$
\le L_1 \|D_{1(n-1)}(t)\| \left| \frac{(1-\alpha)}{M(\alpha)} + \frac{t^\alpha}{M(\alpha)\Gamma(\alpha)} \right|
$$

Consequently, we have

$$
\|D_{1n}(t)\| \le L_1 \left| \frac{(1-\alpha)}{M(\alpha)} + \frac{t^\alpha}{M(\alpha)\Gamma(\alpha)} \right| \|D_{1(n-1)}(t)\|
\tag{26}
$$

In the same manner, the remaining expressions of (26) can be simplified to the following expressions:

$$
\begin{cases}
\|D_{2n}(t)\| \le L_2 \left| \frac{(1-\alpha)}{M(\alpha)} + \frac{t^\alpha}{M(\alpha)\Gamma(\alpha)} \right| \|D_{2(n-1)}(t)\| \\
\|D_{3n}(t)\| \le L_3 \left| \frac{(1-\alpha)}{M(\alpha)} + \frac{t^\alpha}{M(\alpha)\Gamma(\alpha)} \right| \|D_{3(n-1)}(t)\| \\
\|D_{4n}(t)\| \le L_4 \left| \frac{(1-\alpha)}{M(\alpha)} + \frac{t^\alpha}{M(\alpha)\Gamma(\alpha)} \right| \|D_{4(n-1)}(t)\| \\
\|D_{5n}(t)\| \le L_5 \left| \frac{(1-\alpha)}{M(\alpha)} + \frac{t^\alpha}{M(\alpha)\Gamma(\alpha)} \right| \|D_{5(n-1)}(t)\| \\
\|D_{6n}(t)\| \le L_6 \left| \frac{(1-\alpha)}{M(\alpha)} + \frac{t^\alpha}{M(\alpha)\Gamma(\alpha)} \right| \|D_{6(n-1)}(t)\| \\
\|D_{7n}(t)\| \le L_7 \left| \frac{(1-\alpha)}{M(\alpha)} + \frac{t^\alpha}{M(\alpha)\Gamma(\alpha)} \right| \|D_{7(n-1)}(t)\| \\
\|D_{8n}(t)\| \le L_8 \left| \frac{(1-\alpha)}{M(\alpha)} + \frac{t^\alpha}{M(\alpha)\Gamma(\alpha)} \right| \|D_{8(n-1)}(t)\| \\
\|D_{9n}(t)\| \le L_9 \left| \frac{(1-\alpha)}{M(\alpha)} + \frac{t^\alpha}{M(\alpha)\Gamma(\alpha)} \right| \|D_{9(n-1)}(t)\|
\end{cases}
\tag{27}
$$

**Theorem 7**. The mathematical model involving $ABC$ fractional model given in (10) has a solution if we can find $K_0$ satisfying the inequality.

$$
\left( \frac{(1-\alpha)}{M(\alpha)} + \frac{t^\alpha}{M(\alpha)\Gamma(\alpha)} \right) L_i < 1, \quad i = 1,\ 2,\ 3,\ 4,\ 5,\ 6,\ 7,\ 8,\ 9
\tag{28}
$$

**Proof**: From Equation (26) and (27), we have

$$
\begin{cases}
\|D_{1n}(t)\| \le \|S_h(0)\| \left[ \left( \frac{(1-\alpha)}{M(\alpha)} + \frac{t^\alpha}{M(\alpha)\Gamma(\alpha)} \right) L_1 \right]^n \\
\|D_{2n}(t)\| \le \|E_h(0)\| \left[ \left( \frac{(1-\alpha)}{M(\alpha)} + \frac{t^\alpha}{M(\alpha)\Gamma(\alpha)} \right) L_2 \right]^n \\
\|D_{3n}(t)\| \le \|I_h(0)\| \left[ \left( \frac{(1-\alpha)}{M(\alpha)} + \frac{t^\alpha}{M(\alpha)\Gamma(\alpha)} \right) L_3 \right]^n \\
\|D_{4n}(t)\| \le \|T_h(0)\| \left[ \left( \frac{(1-\alpha)}{M(\alpha)} + \frac{t^\alpha}{M(\alpha)\Gamma(\alpha)} \right) L_4 \right]^n \\
\|D_{5n}(t)\| \le \|T_m(0)\| \left[ \left( \frac{(1-\alpha)}{M(\alpha)} + \frac{t^\alpha}{M(\alpha)\Gamma(\alpha)} \right) L_5 \right]^n \\
\|D_{6n}(t)\| \le \|R_h(0)\| \left[ \left( \frac{(1-\alpha)}{M(\alpha)} + \frac{t^\alpha}{M(\alpha)\Gamma(\alpha)} \right) L_6 \right]^n \\
\|D_{7n}(t)\| \le \|S_m(0)\| \left[ \left( \frac{(1-\alpha)}{M(\alpha)} + \frac{t^\alpha}{M(\alpha)\Gamma(\alpha)} \right) L_7 \right]^n \\
\|D_{8n}(t)\| \le \|E_m(0)\| \left[ \left( \frac{(1-\alpha)}{M(\alpha)} + \frac{t^\alpha}{M(\alpha)\Gamma(\alpha)} \right) L_8 \right]^n \\
\|D_{9n}(t)\| \le \|I_m(0)\| \left[ \left( \frac{(1-\alpha)}{M(\alpha)} + \frac{t^\alpha}{M(\alpha)\Gamma(\alpha)} \right) L_9 \right]^n
\end{cases}
\tag{29}
$$

The existence of the solution is verified by theorem 7, and the function $S_h$, $E_h$, $I_h$, $T_h$, $T_m$, $R_h$, $S_m$, $E_m$, $I_m$ are solutions of model (10).

Consider the following conditions are satisfied

$$
\begin{cases}
S_h(t) - S_h(0) = S_{hn}(t) - D_{1n}(t) \\
E_h(t) - E_h(0) = E_{hn}(t) - D_{2n}(t) \\
I_h(t) - I_h(0) = I_{hn}(t) - D_{3n}(t) \\
T_h(t) - T_h(0) = T_{mn}(t) - D_{4n}(t) \\
T_m(t) - T_m(0) = T_{mn}(t) - D_{5n}(t) \\
R_h(t) - R_h(0) = R_{hn}(t) - D_{6n}(t) \\
S_m(t) - S_m(0) = S_{mn}(t) - D_{7n}(t) \\
E_m(t) - E_m(0) = E_{mn}(t) - D_{8n}(t) \\
I_m(t) - I_m(0) = I_{mn}(t) - D_{9n}(t)
\end{cases}
\tag{30}
$$

From Equation (30) we have

$$\|D_{1n}(t)\| \leq \tfrac{(1-\alpha)}{M(\alpha)} \|P_1(\tau, S_{hn}) - P_1(\tau, S_{hn-1})\| + \tfrac{\alpha}{M(\alpha)\Gamma(\alpha)} \int_0^\tau \|P_1(\tau, S_{hn}) - P_1(\tau, S_{hn-1})\| (t-\tau)^{\alpha-1} d\tau$$

$$\leq \tfrac{(1-\alpha)}{M(\alpha)} L_1 \|S_{hn} - S_{hn-1}\| + \tfrac{\alpha^n}{M(\alpha)\Gamma(\alpha)} L_1 \|S_{hn} - S_{hn-1}\|$$

By repeating the process of recursive formula, we have

$$\|D_{1n}(t)\| \leq \left[ \frac{(1-\alpha)}{M(\alpha)} + \frac{t^\alpha}{M(\alpha)\Gamma(\alpha)} \right]^{n+1} (L_1 \|S_{hn} - S_{hn-1}\|)^{n+1}, \tag{31}$$

For $t = K_0$, and Equation (31) becomes

$$\|D_{1n}(t)\| \leq \left[ \frac{(1-\alpha)}{M(\alpha)} + \frac{K_0^\alpha}{M(\alpha)\Gamma(\alpha)} \right]^{n+1} (L_1 \|S_{hn} - S_{hn-1}\|)^{n+1}, \tag{32}$$

Now, by taking the limit of (32), as $n \to \infty$, $\|D_{1n}(t)\| \to 0$, $\left( \frac{(1-\alpha)}{M(\alpha)} + \frac{t^\alpha}{M(\alpha)\Gamma(\alpha)} \right) L_1 < 1$. $\qquad$ (33)

Similarly, we can show that $\|D_{1n}(t)\| \to 0$, $\|D_{2n}(t)\| \to 0$, $\|D_{3n}(t)\| \to 0$, $\|D_{4n}(t)\| \to 0$, $\|D_{5n}(t)\| \to 0$, $\|D_{6n}(t)\| \to 0$, $\|D_{7n}(t)\| \to 0$, $\|D_{8n}(t)\| \to 0$, $\|D_{9n}(t)\| \to 0$, for $\left( \frac{(1-\alpha)}{M(\alpha)} + \frac{t^\alpha}{M(\alpha)\Gamma(\alpha)} \right) L_i < 1, i = 2, 3, 4, 5, 6, 7, 8, 9$.

By using the Banach fixed-point theorem, Theorems 6 and 7 ensure that the solution to model (10) exists.

**Theorem 8**. The *AB* fractional model (10) has a unique solution, provided that

$$\left( \tfrac{1-\alpha}{M(\alpha)} + \tfrac{t^\alpha}{M(\alpha)\Gamma(\alpha)} \right) L_i < 1$$

**Proof**: Assume that $(S_{h1}, E_{h1}, I_{h1}, T_{h1}, T_{m1}, R_{h1}, S_{m1}, E_{m1}, I_{m1})$ are solutions to (10).

Then, $S_h(t) - S_{h1}(t) = \frac{1-\alpha}{M(\alpha)} (P_1(t, S_h) - P_1(t, S_{h1})) + \frac{\alpha}{M(\alpha)\Gamma(\alpha)} \int_0^t (P_1(\tau, S_h) - P_1(\tau, S_{h1})) (t-\tau)^{\alpha-1} d\tau$. By applying the norm on both sides, we obtain

$$\|S_h(t) - S_{h1}(t)\| \leq \tfrac{1-\alpha}{M(\alpha)} L_1 \|S_h - S_{h1}\| + \tfrac{t^\alpha}{M(\alpha)\Gamma(\alpha)} L_1 \|S_h - S_{h1}\|.$$

Since $\left( 1 - L_1 \left( \frac{1-\alpha}{M(\alpha)} + \frac{t^\alpha}{M(\alpha)\Gamma(\alpha)} \right) \right) > 0$, we obtain $\|S_h(t) - S_{h1}(t)\| = 0$. Thus, we have $S_h(t) = S_{h1}(t)$. In similar manner, we can show for the remaining and it completes the proof.

#### 4.2. Malaria-free equilibrium point

The malaria-free equilibrium of the model (10) is given by $M_0 = \left( \frac{\Lambda_h}{\mu}, 0, 0, 0, 0, 0, \frac{\Lambda_m}{\eta}, 0, 0 \right)$

#### 4.3. Basic reproduction number, $\mathcal{R}_0$

To obtain $\mathcal{R}_0$ for system (10), we use the next-generation matrix technique described in [48] and is the spectral radius $\rho(FV^{-1})$, where

$$F = \begin{pmatrix} 0 & 0 & 0 & q\beta_{mh} & 0 \\ 0 & 0 & 0 & 0 & 0 \\ 0 & \frac{q\mu\alpha_{1m}\Lambda_m}{\eta\Lambda_h} & 0 & 0 & \frac{q\mu\alpha_{2m}\Lambda_m}{\eta\Lambda_h} \\ 0 & 0 & 0 & 0 & 0 \\ 0 & 0 & 0 & 0 & 0 \end{pmatrix} \text{ and } V = \begin{pmatrix} (\theta_h + \mu) & 0 & 0 & 0 & 0 \\ -\theta_h & (\delta + \tau_1 + \tau_2 + \mu) & 0 & 0 & 0 \\ 0 & 0 & (\theta_m + \eta) & 0 & 0 \\ 0 & 0 & -\theta_m & \eta & 0 \\ 0 & -\tau_2 & 0 & 0 & (\omega + \gamma + \mu + \psi) \end{pmatrix}$$

By next generation operator method, the basic reproduction number of the model (10) is given as

$$\mathcal{R}_0 = \rho\left(FV^{-1}\right) = \sqrt{\frac{q^2\mu\beta_{hm}\theta_h\theta_m\Lambda_m\left((\gamma+\mu+\omega+\psi)\alpha_{1m}+\tau_2\alpha_{2m}\right)}{\eta^2\Lambda_h\left(\mu+\theta_h\right)\left(\eta+\theta_m\right)\left(\gamma+\mu+\omega+\psi\right)\left(\delta+\mu+\tau_1+\tau_2\right)}} \tag{34}$$

**Theorem 9** [40]. The malaria-free equilibrium point, $M_0$ is locally asymptotically stable if the reproduction number, $\mathcal{R}_0 < 1$ and is unstable if $\mathcal{R}_0 > 1$.

**Proof:** The Jacobian matrix of the model with respect to the state variables at the $M_0$ is as follows:

$$J(M_0) = \begin{bmatrix}
-\mu & 0 & 0 & \rho & 0 & \rho_1 & 0 & 0 & -A \\
0 & -B & 0 & 0 & 0 & 0 & 0 & 0 & A \\
0 & \theta_h & -C & 0 & 0 & \gamma_1 & 0 & 0 & 0 \\
0 & 0 & \tau_1 & -D & \omega & 0 & 0 & 0 & 0 \\
0 & 0 & \tau_2 & 0 & -E & 0 & 0 & 0 & 0 \\
0 & 0 & 0 & 0 & \gamma & -F & 0 & 0 & 0 \\
0 & 0 & -G & 0 & -H & 0 & -\eta & 0 & 0 \\
0 & 0 & G & 0 & H & 0 & 0 & -I & 0 \\
0 & 0 & 0 & 0 & 0 & 0 & 0 & \theta_m & -\eta
\end{bmatrix} \tag{35}$$

Where, $A = q\beta_{mh}$, $B = (\theta_h+\mu)$, $C = (\delta+\tau_1+\tau_2+\mu)$, $D = (\rho+\mu)$, $E = (\omega+\gamma+\mu+\psi)$, $F = (\gamma_1+\rho_1+\mu)$, $G = \frac{q\mu\alpha_{1m}\Lambda_m}{\eta\Lambda_h}$, $H = \frac{q\mu\alpha_{2m}\Lambda_m}{\eta\Lambda_h}$, $I = (\theta_m+\eta)$.

Whose eigenvalues of $\lambda_1 = -\mu$, $\lambda_2 = -\eta$ or the remaining eigenvalues are the roots of the characteristic equation for (35) given by

$$\begin{vmatrix}
-B-\lambda & 0 & 0 & 0 & 0 & 0 & A \\
\theta_h & -C-\lambda & 0 & 0 & \gamma_1 & 0 & 0 \\
0 & \tau_1 & -D-\lambda & \omega & 0 & 0 & 0 \\
0 & \tau_2 & 0 & -E-\lambda & 0 & 0 & 0 \\
0 & 0 & 0 & \gamma & -F-\lambda & 0 & 0 \\
0 & G & 0 & H & 0 & -I-\lambda & 0 \\
0 & 0 & 0 & 0 & 0 & \theta_m & -\eta-\lambda
\end{vmatrix} = 0 \tag{36}$$

$$(-D-\lambda)\left(-A(F+\lambda)\theta_h\theta_m\left(G(E+\lambda)+H\tau_2\right)+(I+\lambda)(B+\lambda)(\eta+\lambda)\left((C+\lambda)(E+\lambda)(F+\lambda)-\gamma\gamma_1\tau_2\right)\right) = 0 \tag{37}$$

We have $\lambda_1 = -\mu$, $\lambda_2 = -\eta$, $\lambda_3 = -(\rho+\mu)$ and the remaining eigenvalues are obtained from equation

$$P_6\lambda^6 + P_5\lambda^5 + P_4\lambda^4 + P_3\lambda^3 + P_2\lambda^2 + P_1\lambda + P_0 = 0 \tag{38}$$

Where, $P_6 = 1 > 0$, $P_5 = (\gamma+\delta+2\eta+4\mu+\omega+\gamma_1+\theta_h+\theta_m+\rho_1+\tau_1+\tau_2) > 0$, $P_4 > 0$, $P_3 > 0$, $P_2 > 0$, $P_1 = \left(IBCEF+IBCF\eta+IBEF\eta+ICEF\eta+BCEF\eta+IBCE\eta\left(1-\mathcal{R}_0^2\right)-AFG\theta_h\theta_m-(IB+I\eta+B\eta)\gamma\gamma_1\tau_2\right)$ and $P_0 = IB\eta\left(CEF\left(1-\mathcal{R}_0^2\right)-\gamma\gamma_1\tau_2\right) > 0$, where $\mathcal{R}_0 < 1$ and $\gamma\gamma_1\tau_2 < CEF$.

Applying the Routh-Hurwitz stability criterion [34] and after some little algebraic manipulations, it can be shown that the eigenvalues of the block matrix have negative real parts. If $\mathcal{R}_0 > 1$, then $P_1 < 0$, thus the matrix $JM_0$ has at least one eigenvalue with positive real part. Hence, malaria-free equilibrium is locally asymptotically stable if $\mathcal{R}_0 < 1$ and unstable if $\mathcal{R}_0 > 1$.

**Theorem 10**. If $\mathcal{R}_0 > 1$, then the model (10) has unique malaria present equilibrium $M^* = {S_h}^*, {E_h}^*, {I_h}^*, {T_h}^*, {T_m}^*, {R_h}^*, {S_m}^*, {E_m}^*, {I_m}^*$, where

$$S_h^* = \frac{(\theta_h + \mu)}{\lambda_h}\left(\frac{(\delta + \tau_1 + \tau_2 + \mu)}{\theta_h} - \frac{\gamma_1\gamma\tau_2}{\theta_h\,(\gamma_1 + \rho_1 + \mu)\,(\omega + \gamma + \mu + \psi)}\right){I_h}^*,\quad E_h^* = \left(\frac{(\delta + \tau_1 + \tau_2 + \mu)}{\theta_h} - \frac{\gamma_1\gamma\tau_2}{\theta_h\,(\gamma_1 + \rho_1 + \mu)\,(\omega + \gamma + \mu + \psi)}\right){I_h}^*,$$

$$T_h^* = \left(\frac{\tau_1}{(\rho + \mu)} + \frac{\omega\tau_2}{(\omega + \gamma + \mu + \psi)}\right){I_h}^*,\quad T_m^* = \left(\frac{\tau_2}{(\omega + \gamma + \mu + \psi)}\right){I_h}^*,\quad R_h^* = \left(\frac{\gamma\tau_2}{(\gamma_1 + \rho_1 + \mu)\,(\omega + \gamma + \mu + \psi)}\right){I_h}^*,$$

$$I_h^* = -\frac{\Lambda_h}{\frac{\rho\tau_1}{\mu + \rho} + \frac{\rho\omega\tau_2}{\gamma + \mu + \psi + \omega} + \frac{\gamma\rho_1\tau_2}{(\gamma + \mu + \psi + \omega)(\mu + \gamma_1 + \rho_1)} + \frac{(-\mu - \theta_h)(\mu + \lambda_h)\left(\delta + \mu + \tau_1 + \left(1 - \frac{\gamma\gamma_1}{(\gamma + \mu + \psi + \omega)(\mu + \gamma_1 + \rho_1)}\right)\tau_2\right)}{\theta_h\lambda_h}},\quad S_m^* = \left(\frac{\Lambda_m}{(\lambda_m + \eta)}\right),$$

$$E_m^* = \left(\frac{\Lambda_m\lambda_m}{(\lambda_m + \eta)(\theta_m + \eta)}\right)\text{ and }\quad I_m^* = \left(\frac{\theta_m\lambda_m\Lambda_m}{(\theta_m + \eta)(\lambda_m + \eta)}\right).$$

**Proof.** Let ${\lambda_h}^* = q\beta_{mh}\frac{{I_m}^*}{N_h}$ be the force of infection for humans and ${\lambda_m}^* = q\alpha_{1m}\frac{{I_h}^*}{N_h} + q\alpha_{2m}\frac{{T_m}^*}{N_h}$ be the force of infection for mosquitoes. Then, by substituting ${I_h}^*$ and ${I_m}^*$ in the ${\lambda_h}^*$ and ${\lambda_m}^*$, respectively, we get the simplified form

$$\lambda_h^* = \frac{q\mu\left(-\alpha_{1m} - \frac{\alpha_{2m}\tau_2}{\gamma + \mu + \psi + \omega}\right)}{\frac{\rho\tau_1}{\mu + \rho} + \frac{\rho\omega\tau_2}{\gamma + \mu + \psi + \omega} + \frac{\gamma\rho_1\tau_2}{(\gamma + \mu + \psi + \omega)(\mu + \gamma_1 + \rho_1)} + \frac{(-\mu - \theta_h)(\mu + \lambda_m)(\delta + \mu + \tau_1 + (1 - \frac{\gamma\gamma_1}{(\gamma + \mu + \psi + \omega)(\mu + \gamma_1 + \rho_1)})\tau_2)}{\theta_h\lambda_m}} \tag{39}$$

$$P_1\lambda_h + P_0 = 0 \tag{40}$$

Where, $P_1 = (IBDE\Lambda_h\,(FC - \gamma\gamma_1\tau_2) + A\theta_m\Lambda_m\,(BDEF(\delta + \mu) + EF\,(BD - \rho\theta_h)\,\tau_1 + D\,(BEF - BE\gamma\gamma_1 - \theta_h\,(F\rho\omega + \gamma\rho_1))\,\tau_2))$ and $P_0 = D\eta^2\Lambda_h IB(CEF\,(\mathcal{R}_0^2 - 1) + \gamma\gamma_1\tau_2)$.

Hence, we have established the following result.

**Theorem 11.** The model (10) admits precisely [40].

a. One endemic equilibrium point if $\mathcal{R}_0 > 1$ and $P_1 < 0$ or $P_0 > 0$ and $P_1 < 0$.

b. One endemic equilibrium point if $\mathcal{R}_0 < 1$, $\gamma\gamma_1\tau_2 > CEF$ and $P_1 < 0$ or $P_0 > 0$ and $P_1 < 0$.

c. One endemic equilibrium point if $\mathcal{R}_0 < 1$, $\gamma\gamma_1\tau_2 < CEF$ and $P_1 > 0$ or $P_0 < 0$ and $P_1 > 0$.

d. No equilibrium point otherwise.

**Theorem 12.** The malaria-present equilibrium, $M^* = \left({S_h}^*,\ {E_h}^*, {I_h}^*, {T_h}^*, {T_m}^*, {R_h}^*, {S_m}^*, {E_m}^*, {I_m}^*\right)$ of the model (10), is locally asymptotically stable (LAS), if $\mathcal{R}_0 > 1$ and unstable if $\mathcal{R}_0 < 1$.

**Proof**: The Jacobian matrix of the model (10) is obtained by:

$$J(E^*) = \begin{pmatrix} -a_1 & 0 & 0 & a_4 & 0 & a_6 & 0 & 0 & -a_9 \\ b_1 & -b_2 & 0 & 0 & 0 & 0 & 0 & 0 & b_9 \\ 0 & c_2 & -c_3 & 0 & 0 & c_6 & 0 & 0 & 0 \\ 0 & 0 & d_3 & -d_4 & d_5 & 0 & 0 & 0 & 0 \\ 0 & 0 & e_3 & 0 & -e_5 & 0 & 0 & 0 & 0 \\ 0 & 0 & 0 & 0 & f_5 & -f_6 & 0 & 0 & 0 \\ 0 & 0 & -g_3 & 0 & -g_5 & 0 & -g_7 & 0 & 0 \\ 0 & 0 & h_3 & 0 & h_5 & 0 & h_7 & -h_8 & 0 \\ 0 & 0 & 0 & 0 & 0 & 0 & 0 & k_8 & -k_9 \end{pmatrix} \tag{41}$$

Where, $a_1 = \mu + \frac{q\beta_{mh}{I_m}^*}{{N_h}^*}$, $a_4 = \rho$, $a_6 = \rho_1$, $a_9 = \frac{q\beta_{mh}{S_h}^*}{{N_h}^*}$, $b_1 = \frac{q\beta_{mh}{I_m}^*}{{N_h}^*}$, $b_2 = \theta_h + \mu$, $b_9 = \frac{q\beta_{mh}{S_h}^*}{{N_h}^*}$, $c_2 = \theta_h$, $c_3 = (\delta + \tau_1 + \tau_2 + \mu)$, $c_6 = \gamma_1$, $d_3 = \tau_1$, $d_4 = (\rho + \mu)$, $d_5 = \omega$, $e_1 = e_2 = e_3 = \tau_2$, $e_5 = (\omega + \gamma + \mu + \psi)$, $f_5 = \gamma$, $f_6 = \gamma_1 + \rho_1 + \mu$, $g_3 = \frac{q\alpha_{1m}}{{N_h}^*}{S_m}^*$, $g_5 = \frac{q\alpha_{2m}}{{N_h}^*}{S_m}^*$, $g_7 = (q\alpha_{1m}\frac{{I_h}^*}{{N_h}^*} + q\alpha_{2m}\frac{{T_S}^*}{{N_h}^*} + \eta)$, $h_3 = \frac{q\alpha_{1m}}{{N_h}^*}{S_m}^*$, $h_5 = \frac{q\alpha_{2m}}{{N_h}^*}{S_m}^*$, $h_7 = (q\alpha_{1m}\frac{{I_h}^*}{{N_h}^*} + q\alpha_{2m}\frac{{T_S}^*}{{N_h}^*})$, $h_8 = \theta_m + \eta$, $k_8 = \theta_m$, $k_9 = \eta$.

The corresponding characteristic equation of the Jacobian matrix with eigenvalue $\lambda$ is given by $\left| J(E^*) - \lambda I \right| = 0$; that is,

$$P_9\lambda^9 + P_8\lambda^8 + P_7\lambda^7 + P_6\lambda^6 + P_5\lambda^5 + P_4\lambda^4 + P_3\lambda^3 + P_2\lambda^2 + P_1\lambda + P_0 = 0 \tag{42}$$

Where, $P_0, P_1, P_2, \ldots P_9$ and proof of Routh-Hurwitz stability criterion see (Appendix 1).

Thus, we show that when $\mathcal{R}_0 > 1$, all the coefficients $P_i$ of the characteristic Equation (42), and the first column values $b_1, c_1, d_1, e_1, f_1, g_1, h_1$ and $i_0$ of the Routh array are positive, so by the Routh-Hurwitz stability criterion, all the eigenvalues of the Jacobian matrix (41) have negative real parts. Thus, the malaria-present equilibrium point is locally asymptotically stable for $\mathcal{R}_0 > 1$.

## 5. Sensitivity analysis

This section conducts a sensitivity analysis of the basic reproductive number, crucial for designing mitigation strategies to slow malaria spread. It aids researchers, public health officials, and policymakers in prioritizing interventions based on influencing factors and understanding the effects of each parameter on reproduction numbers [49,50]. The sensitivity analysis of the basic reproduction number, $\mathcal{R}_0$ provides insight into how variations in different parameters influence the spread of malaria. The sign of the sensitivity index indicates whether an increase in the parameter value would increase or decrease, $\mathcal{R}_0$.

**Definition 1:** Normalized forward sensitivity index of $\mathcal{R}_0$ which is differentiable with respect to a given parameter $p$ is defined as [51].

$$SI(p) = \left(\frac{\partial \mathcal{R}_0}{\partial p}\right) * \left(\frac{p}{\mathcal{R}_0}\right) \tag{43}$$

Using definition 1 and parameter values given in Table 2, the respective sensitivity indices values for reproduction number are computed in Table 3 and we plot the sensitivity indices in Fig 2.

The sensitivity index with positive values increases the basic reproduction number, $\mathcal{R}_0$ when they are increased, while the negative values decrease $\mathcal{R}_0$ when they are increased. From Fig 3 and the parameter values displayed in Table 3, we can observe that factors such as $q$, $\beta_{mh}$, $\Lambda_m$, $\alpha_{2m}$, $\eta$, $\tau_1$ and $\omega$ are the most sensitive parameters. The basic reproduction number, $\mathcal{R}_0$ is impacted positively by the parameters $q$, $\beta_{mh}$, $\Lambda_m$, and $\alpha_{2m}$, which means that decreasing their value will cause a decrease in $\mathcal{R}_0$, while negatively impacted by the parameters $\eta$, $\tau_1$ and $\omega$, which means that increasing this parameter will cause a decrease in $\mathcal{R}_0$. This highlights the importance of targeting these parameters for effective control measures.

Table 3. Sensitivity indices of $\mathcal{R}_0$ to parameters evaluated at the parameter values given in Table 1.

| Parameter | Sensitivity Index | Sign | Parameter | Sensitivity Index | Sign |
|---|---|---|---|---|---|
| $\Lambda_h$ | −0.5 | −ve | $\mu$ | 0.008 | +ve |
| $\Lambda_m$ | 0.5 | +ve | $\eta$ | −0.36 | −ve |
| $\beta_{mh}$ | 0.5 | +ve | $\tau_1$ | −0.31 | −ve |
| $\alpha_{1m}$ | 0.1 | +ve | $\tau_2$ | −0.2 | −ve |
| $\alpha_{2m}$ | 0.36 | +ve | $\psi$ | −0.003 | −ve |
| $\delta$ | −0.001 | −ve | $q$ | 1 | +ve |
| $\theta_h$ | 0.0002 | +ve | $\omega$ | −0.3 | −ve |
| $\theta_m$ | 0.19 | +ve | $\gamma$ | −0.06 | −ve |

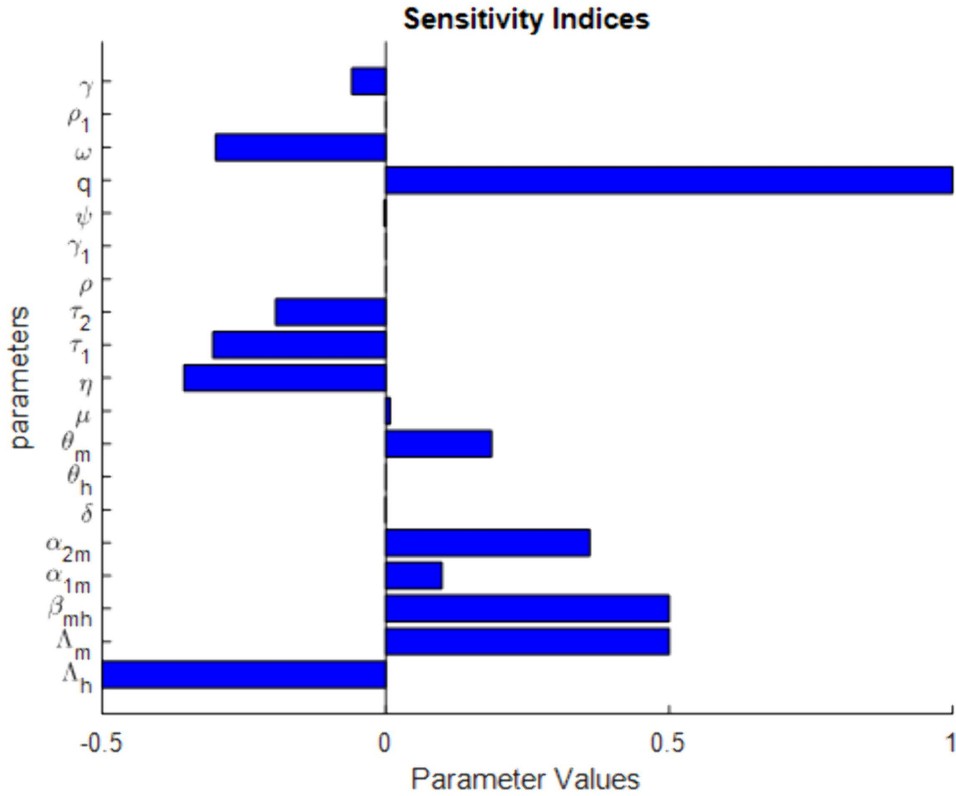

**Fig 2. Sensitivity indices of the model parameters for $\mathcal{R}_0$.**

## 6. Numerical solution of fractional-order model

In this section, we present the numerical simulation of the model system (10). Using the technique developed by Toufik and Atangana's numerical scheme [52]. The numerical method used combines the two-step Lagrange polynomial and the fundamental theorem of fractional calculus. A computer software, MATLAB R2023a, is used for all the simulation results obtained in this study. The initial population used for the numerical simulation is $S_h(0) = 2000$, $E_h(0) = 1000$, $I_h(0) = 500$, $T_h(0) = 300$, $T_m(0) = 500$, $R_h(0) = 400$, $S_m(0) = 8000$, $E_m(0) = 5000$, $I_m(0) = 2000$, and the parameter values are given in Table 2.

Consider nonlinear fractional differential equations ${}^{ABC}_0D^\alpha_t X(t) = P_i(t, X(t))$, $X(0) = X_0$,

Applying the fundamental theorem of fractional calculus, we have a fractional integral equation:

$$X(t) - X(0) = \frac{(1-\alpha)}{M(\alpha)} P_i(t, X(t)) + \frac{\alpha}{M(\alpha)\Gamma(\alpha)} \int_0^t P_i(\tau, X(\tau))(t-\tau)^{\alpha-1} d\tau \tag{44}$$

At a given point $t = t_{n+1}$ and $n = 0, 1, 2, 3 \ldots$, Equation (44) reformulated as discussed in [52].

$$X(t_{n+1}) - X(0) = \frac{(1-\alpha)}{M(\alpha)} P_i(t_n, X(t_n)) + \frac{\alpha}{M(\alpha)\Gamma(\alpha)} \int_0^{t_{n+1}} P_i(\tau, X(\tau))(t_{n+1}-\tau)^{\alpha-1} d\tau.$$

$$X(t_{n+1}) = X(0) + \frac{(1-\alpha)}{M(\alpha)} P_i(t_n, X(t_n)) + \frac{\alpha}{M(\alpha)\Gamma(\alpha)} \sum_{j=0}^n \int_{t_j}^{t_{j+1}} P_i(\tau, X(\tau))(t_{n+1}-\tau)^{\alpha-1} d\tau \tag{45}$$

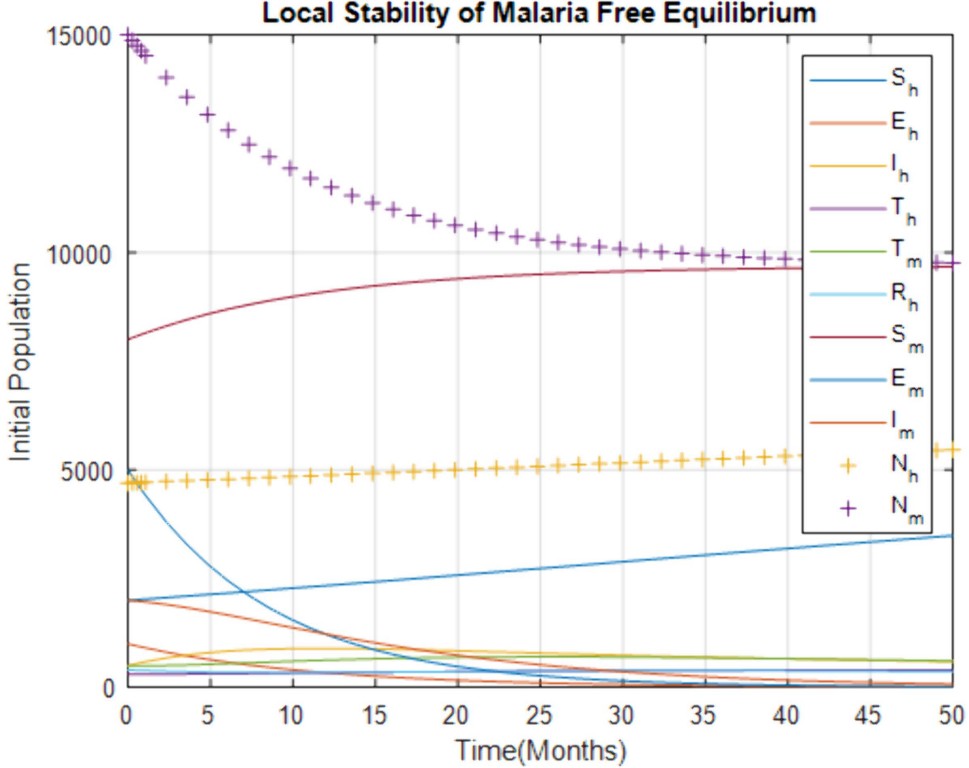

**Fig 3. Local stability of malaria-free equilibrium points.**

Within the interval $[t_j, t_{j+1}]$, the function $P_i(\tau, X(\tau))$, using the two-step Lagrange polynomial interpolation, can be approximate as follows:

$$P_i(\tau, X(\tau)) \cong I_k(\tau) = \frac{(\tau - t_{j-1})}{t_j - t_{j-1}} P_i(t_j, X(t_j)) - \frac{(\tau - t_j)}{t_j - t_{j-1}} P_i(t_{j-1}, X(t_{j-1}))$$

$$= \frac{P_i(t_j, X(t_j))}{h}(\tau - t_{j-1}) - \frac{P_i(t_{j-1}, X(t_{j-1}))}{h}(\tau - t_j) \tag{46}$$

The equation (46) approximation can therefore be included in equation takes the form:

$$X(t_{n+1}) = \frac{(1-\alpha)}{M(\alpha)} P_i(t_n, X(t_n)) + \frac{\alpha}{M(\alpha)\Gamma(\alpha)} \sum_{j=0}^{n} \left( \frac{P_i(t_j, X(t_j))}{h} \int_{t_j}^{t_{j+1}} (\tau - t_{j-1})(t_{n+1} - \tau)^{\alpha-1} d\tau \right.$$

$$\left. - \frac{P_i(t_{j-1}, X(t_{j-1}))}{h} \int_{t_j}^{t_{j+1}} (\tau - t_j)(t_{n+1} - \tau)^{\alpha-1} d\tau \right) \tag{47}$$

Solving the integrals of Equation (47), we obtain.

$$X(t_{n+1}) = X(t_0) + \frac{(1-\alpha)}{M(\alpha)} P_i(t_n, X(t_n)) + \frac{\alpha}{M(\alpha)} \sum_{j=0}^{n} \left( \frac{h^\alpha P_i(t_j, X(t_j))}{\Gamma(\alpha+2)} [(n+1-j)^\alpha(n-j+2+\alpha) \right.$$

$$\left. - (n-j)^\alpha(n-j+2+2\alpha)] - \frac{h^\alpha P_i(t_{j-1}, X(t_{j-1}))}{\Gamma(\alpha+2)} [(n+1-j)^{\alpha+1} - (n-j)^\alpha(n-j+1+\alpha)] \right) \tag{48}$$

By adopting the numerical scheme (48) into the proposed malaria model Equation (10) yields the following numerical solution:

$$S_h(t_{n+1}) = S_h(t_0) + \frac{(1-\alpha)}{M(\alpha)}P_1(t_n, S_h(t_n)) + \frac{\alpha}{M(\alpha)}\sum_{j=0}^{n}\left(\frac{h^\alpha P_1(t_j, S_h(t_j))}{\Gamma(\alpha+2)}\pi_1 - \frac{h^\alpha P_1(t_{j-1}, S_h(t_{j-1}))}{\Gamma(\alpha+2)}\pi_2\right) \quad (49)$$

$$E_h(t_{n+1}) = E_h(t_0) + \frac{(1-\alpha)}{M(\alpha)}P_2(t_n, E_h(t_n)) + \frac{\alpha}{M(\alpha)}\sum_{j=0}^{n}\left(\frac{h^\alpha P_2(t_j, E_h(t_j))}{\Gamma(\alpha+2)}\pi_1 - \frac{h^\alpha P_2(t_{j-1}, E_h(t_{j-1}))}{\Gamma(\alpha+2)}\pi_2\right) \quad (50)$$

$$I_h(t_{n+1}) = I_h(t_0) + \frac{(1-\alpha)}{M(\alpha)}P_3(t_n, I_h(t_n)) + \frac{\alpha}{M(\alpha)}\sum_{j=0}^{n}\left(\frac{h^\alpha P_3(t_j, I_h(t_j))}{\Gamma(\alpha+2)}\pi_1 - \frac{h^\alpha P_3(t_{j-1}, I_h(t_{j-1}))}{\Gamma(\alpha+2)}\pi_2\right) \quad (51)$$

$$T_h(t_{n+1}) = T_h(t_0) + \frac{(1-\alpha)}{M(\alpha)}P_4(t_n, T_h(t_n)) + \frac{\alpha}{M(\alpha)}\sum_{j=0}^{n}\left(\frac{h^\alpha P_4(t_j, T_h(t_j))}{\Gamma(\alpha+2)}\pi_1 - \frac{h^\alpha P_4(t_{j-1}, T_h(t_{j-1}))}{\Gamma(\alpha+2)}\pi_2\right) \quad (52)$$

$$T_m(t_{n+1}) = T_m(t_0) + \frac{(1-\alpha)}{M(\alpha)}P_5(t_n, T_m(t_n)) + \frac{\alpha}{M(\alpha)}\sum_{j=0}^{n}\left(\frac{h^\alpha P_5(t_j, T_m(t_j))}{\Gamma(\alpha+2)}\pi_1 - \frac{h^\alpha P_5(t_{j-1}, T_m(t_{j-1}))}{\Gamma(\alpha+2)}\pi_2\right) \quad (53)$$

$$R_h(t_{n+1}) = R_h(t_0) + \frac{(1-\alpha)}{M(\alpha)}P_6(t_n, R_h(t_n)) + \frac{\alpha}{M(\alpha)}\sum_{j=0}^{n}\left(\frac{h^\alpha P_6(t_j, R_h(t_j))}{\Gamma(\alpha+2)}\pi_1 - \frac{h^\alpha P_6(t_{j-1}, R_h(t_{j-1}))}{\Gamma(\alpha+2)}\pi_2\right) \quad (54)$$

$$S_m(t_{n+1}) = S_m(t_0) + \frac{(1-\alpha)}{M(\alpha)}P_7(t_n, S_m(t_n)) + \frac{\alpha}{M(\alpha)}\sum_{j=0}^{n}\left(\frac{h^\alpha P_7(t_j, S_m(t_j))}{\Gamma(\alpha+2)}\pi_1 - \frac{h^\alpha P_7(t_{j-1}, S_m(t_{j-1}))}{\Gamma(\alpha+2)}\pi_2\right) \quad (55)$$

$$E_m(t_{n+1}) = E_m(t_0) + \frac{(1-\alpha)}{M(\alpha)}P_8(t_n, E_m(t_n)) + \frac{\alpha}{M(\alpha)}\sum_{j=0}^{n}\left(\frac{h^\alpha P_8(t_j, E_m(t_j))}{\Gamma(\alpha+2)}\pi_1 - \frac{h^\alpha P_8(t_{j-1}, E_m(t_{j-1}))}{\Gamma(\alpha+2)}\pi_2\right) \quad (56)$$

$$I_m(t_{n+1}) = I_m(t_0) + \frac{(1-\alpha)}{M(\alpha)}P_9(t_n, I_m(t_n)) + \frac{\alpha}{M(\alpha)}\sum_{j=0}^{n}\left(\frac{h^\alpha P_9(t_j, I_m(t_j))}{\Gamma(\alpha+2)}\pi_1 - \frac{h^\alpha P_9(t_{j-1}, I_m(t_{j-1}))}{\Gamma(\alpha+2)}\pi_2\right) \quad (57)$$

Where $\pi_1 = ((n+1-j)^\alpha(n-j+2+\alpha) - (n-j)^\alpha(n-j+2+2\alpha))$ and $\pi_2 = ((n+1-j)^{\alpha+1} - (n-j)^\alpha(n-j+1+\alpha))$.

In Fig 4, the *MFE* is analyzed to determine its stability using the basic reproduction number, $\mathcal{R}_0$. If $\mathcal{R}_0 < 1$, the disease cannot sustain itself within the population, and the *MFE* is considered locally stable. This means that any small perturbation or introduction of a few infected individuals will not lead to a widespread outbreak, as the infection will eventually die out. The figure likely shows that when the parameters are set such that $\mathcal{R}_0 < 1$, the populations of infected humans and mosquitoes decline over time, approaching zero. This indicates that the *MFE* is stable and that public health measures that reduce $\mathcal{R}_0$ below one are effective in controlling the spread of malaria.

Fig 5 illustrates the dynamics of the human population when the basic reproduction number $\mathcal{R}_0$, is greater than 1, indicating that each infected individual is spreading the disease to more than one other person. The susceptible human and mosquito population declines rapidly, showing that the disease is spreading quickly through the population. This underscores the need for interventions to reduce $\mathcal{R}_0$, such as mosquito control measures and effective treatment strategies.

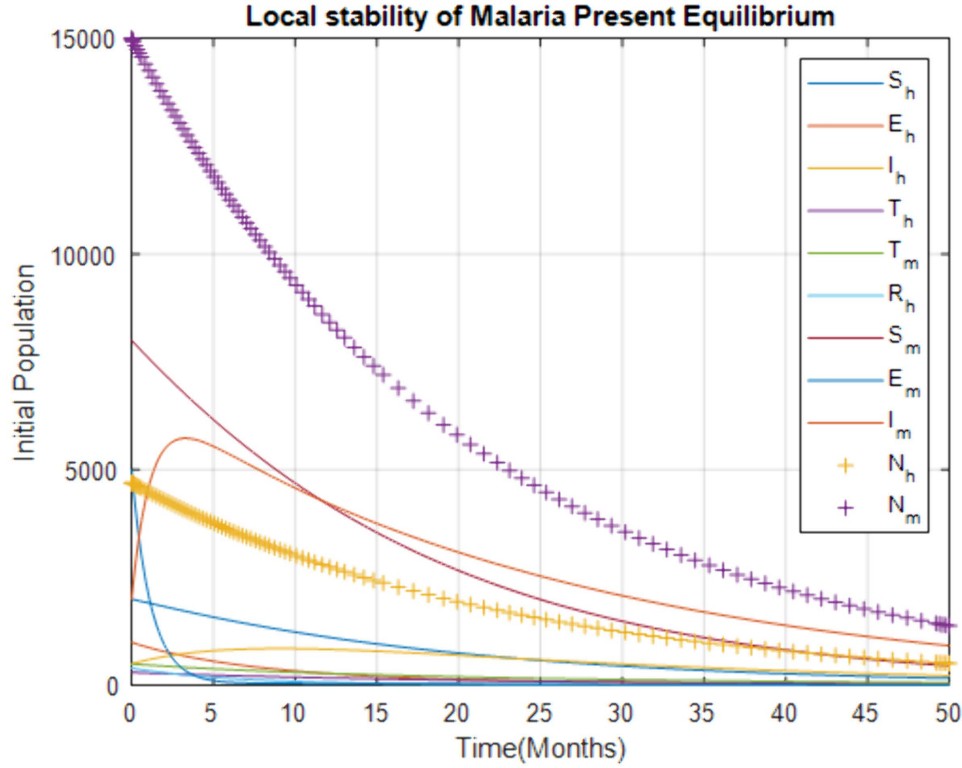

**Fig 4. Local stability of malaria-present equilibrium points.**

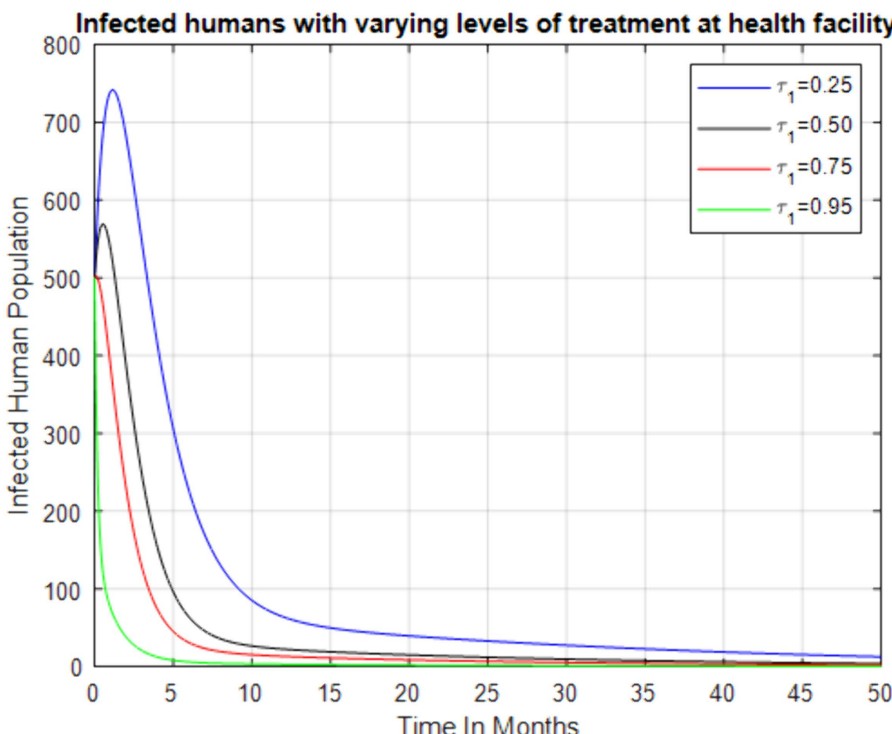

**Fig 5. Effect of increasing treatment rate at health facilities on the infected human.**

Fig 6 shows the effect of increasing the treatment rate at health facilities on the infected human population. As the treatment rate at health facilities increases, the infected human population initially rises slightly but then decreases over time. This indicates that increasing access to and utilization of professional health facilities effectively reduces the infected population, leading to the eventual elimination of the disease in a relatively short period. The figure highlights the importance of strengthening health facility infrastructure and encouraging people to seek treatment in professional settings.

Fig 7 shows that as the use of traditional medicine increases, the infected population also rises for a few months before gradually declining. However, the time required to eliminate the disease is much longer (over 50 months) compared to professional health facilities. This suggests that while traditional medicine might offer some relief, it is significantly less effective in eradicating the disease and may prolong the disease's presence in the community.

Fig 8 demonstrates the effect of increasing the rate at which individuals move from traditional medicine treatment to health facilities due to the ineffectiveness of traditional treatments (like inappropriate dosage, the failure to combine medications, and the presence of counterfeit drugs that contain low doses of the drug, i.e., this may causes anti-malarial drug resistance [36]). As the rate of progression to health facilities increases, the population of individuals using traditional medicine initially rises but then decreases significantly. This indicates that ineffective traditional treatments can delay proper care and lead to the worsening of the disease.

Fig 9 likely shows how increasing the rate of treatment at health facilities ($\tau_1$) reduces the basic reproduction number $\mathcal{R}_0$. A critical intersection occurs at $\mathcal{R}_0 = 1$ and $\tau_1 = 0.17$, indicating a threshold where the spread of the infection is balanced by the rate of treatment. Achieving a treatment rate above this threshold is essential for reducing the reproduction number below 1, effectively controlling the disease. The figure emphasizes the effectiveness of professional medical treatment in reducing the overall transmission of malaria.

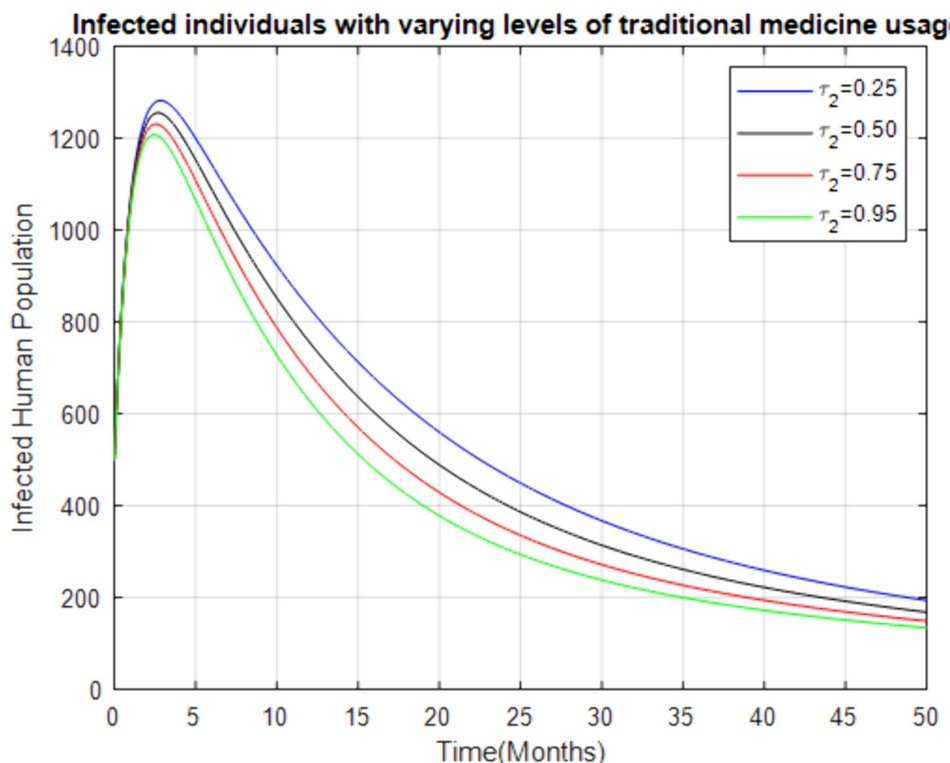

**Fig 6. Effect of increasing treatment rate of traditional medicine usage on the infected human.**

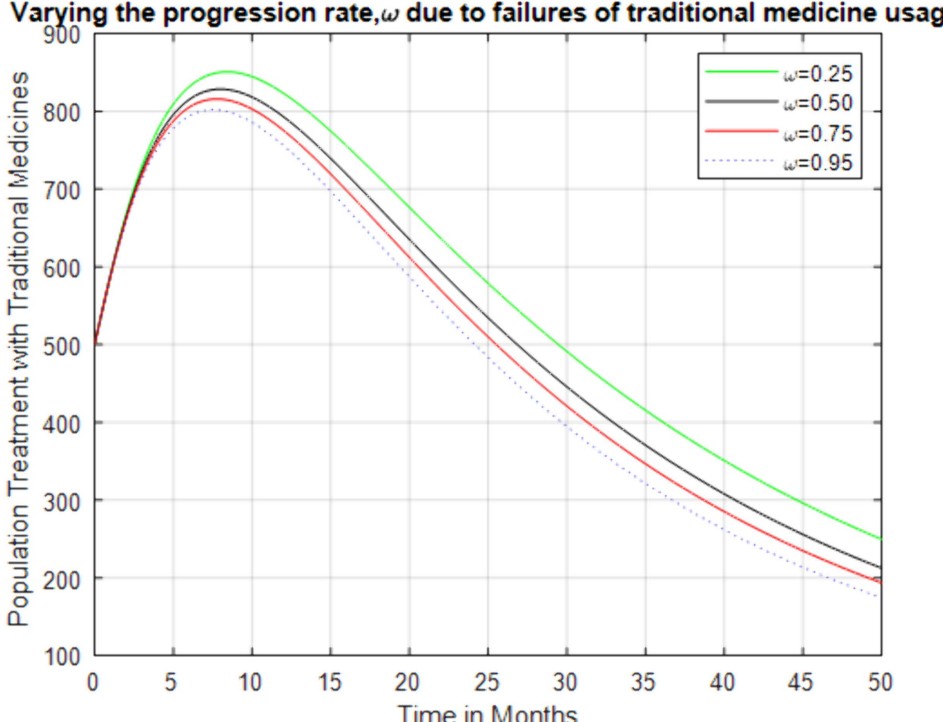

**Fig 7. Effect of increasing progression rate, $\omega$ on individual treatment with traditional medicine class.**

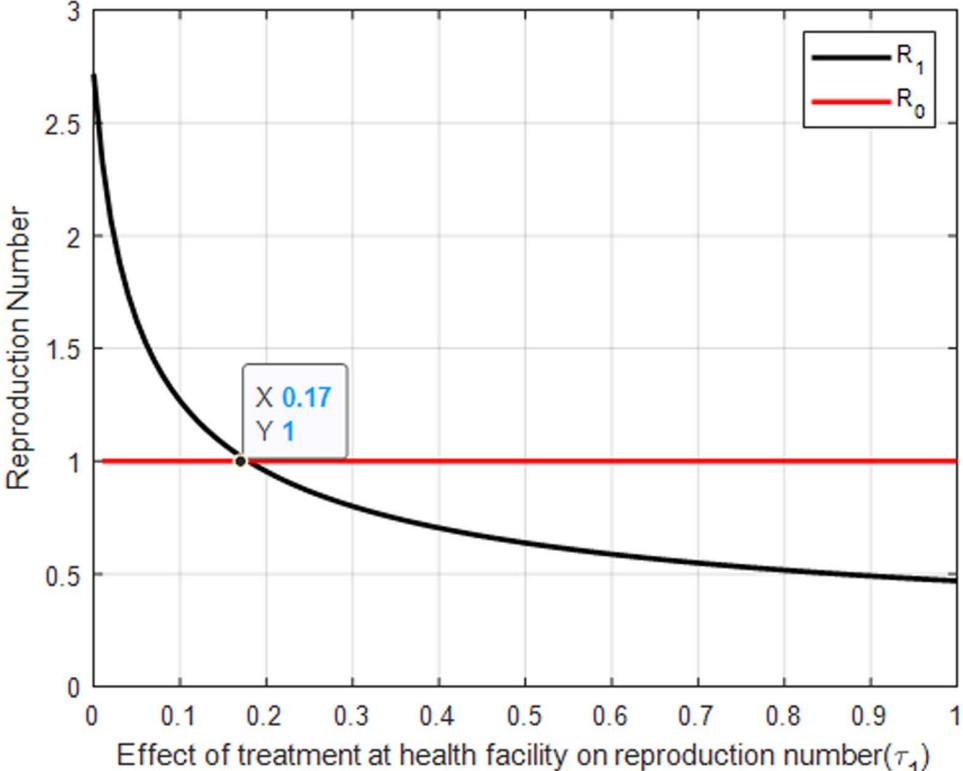

**Fig 8. The effect of treatment at health facilities on reproduction number.**

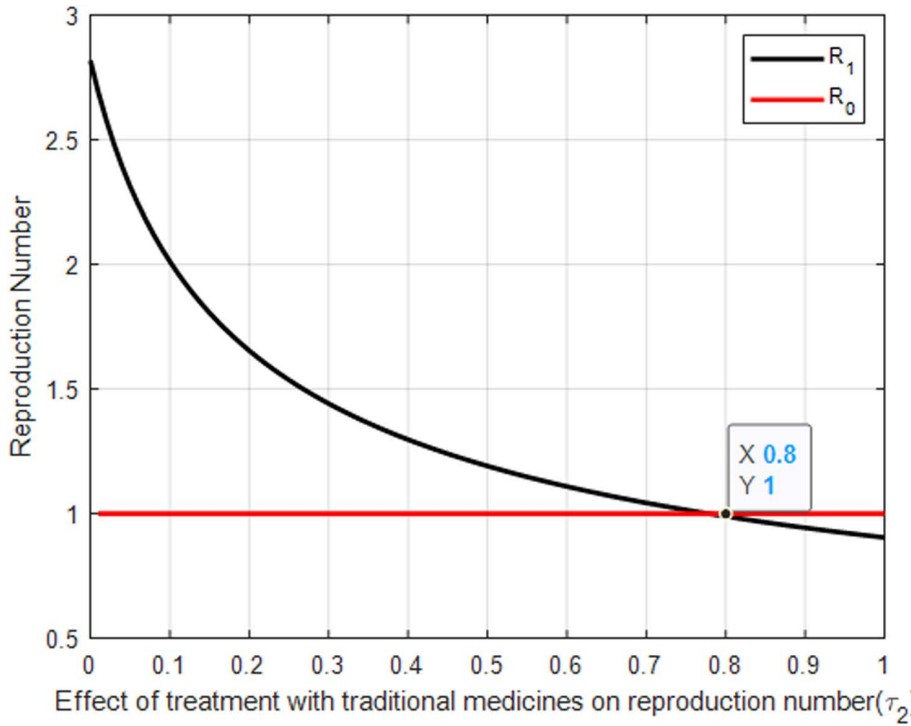

**Fig 9. The effect of treatment with traditional medicines on reproduction number.**

Fig 9 explores how traditional medicine treatment rate ($\tau_2$) affects the basic reproduction number $\mathcal{R}_0$. It likely shows that reliance on traditional medicine alone is less effective at reducing $\mathcal{R}_0$, compared to professional health facility treatments. This underscores the limited impact of traditional remedies on controlling the spread of malaria, the need for integrating modern medical practices into malaria treatment strategies and public health efforts should focus on encouraging the use of clinically validated treatments.

Fig 10 depicts that as the fractional order $\alpha$ increases, the susceptible human population increases slowly over time. A higher fractional order results in a faster increase of the susceptible population, indicating a lower rate of individuals becoming exposed or infected.

Fig 11 illustrates that the exposed human population increases more rapidly with lower fractional orders, indicating that the transition from susceptible to exposed is less aggressive.

Fig 12 shows that higher fractional orders result in a more significant increase in the infected population, which then decreases over time. Higher fractional orders correlate with a rapid increase in infections, followed by a decline, potentially due to increased treatment or death rates.

Fig 13 shows the population receiving treatment at health facilities increases with higher fractional orders and eventually stabilizes which suggests a balance between new infections and treatment. As more individuals seek treatment at health facilities, the infected population decreases, reflecting the effectiveness of professional healthcare interventions.

Fig 14 represents the population receiving treatment with traditional medicines. The trend is similar to those undergoing health facility treatments but may indicate less effective outcomes. This suggests that TMs may offer temporary relief, but may not be as effective in the long term compared to professional health facility treatments.

Fig 15 shows that the recovered population increases with higher fractional orders, showing a positive trend towards recovery over time. This suggests that more individuals are recovering, possibly indicating the effectiveness of both

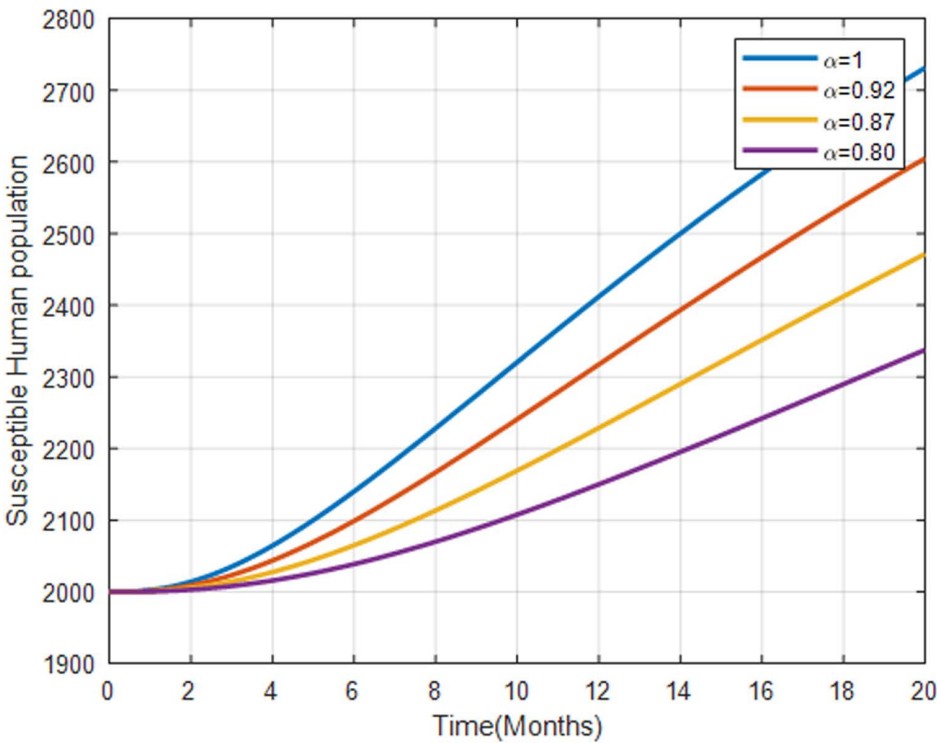

**Fig 10. Susceptible human population with fractional order $\alpha = 0.80$, $\alpha = 0.87$, $\alpha = 0.92$ and.$\alpha = 1$.**

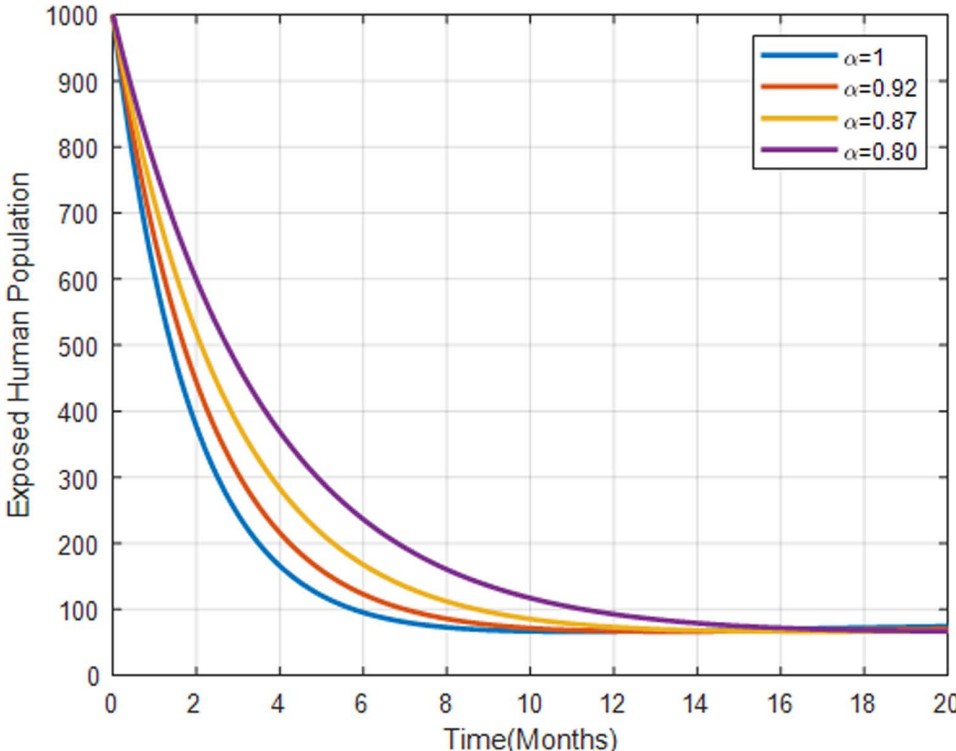

**Fig 11. Exposed human population with fractional order $\alpha = 0.80$, $\alpha = 0.87$, $\alpha = 0.92$ and $\alpha = 1$.**

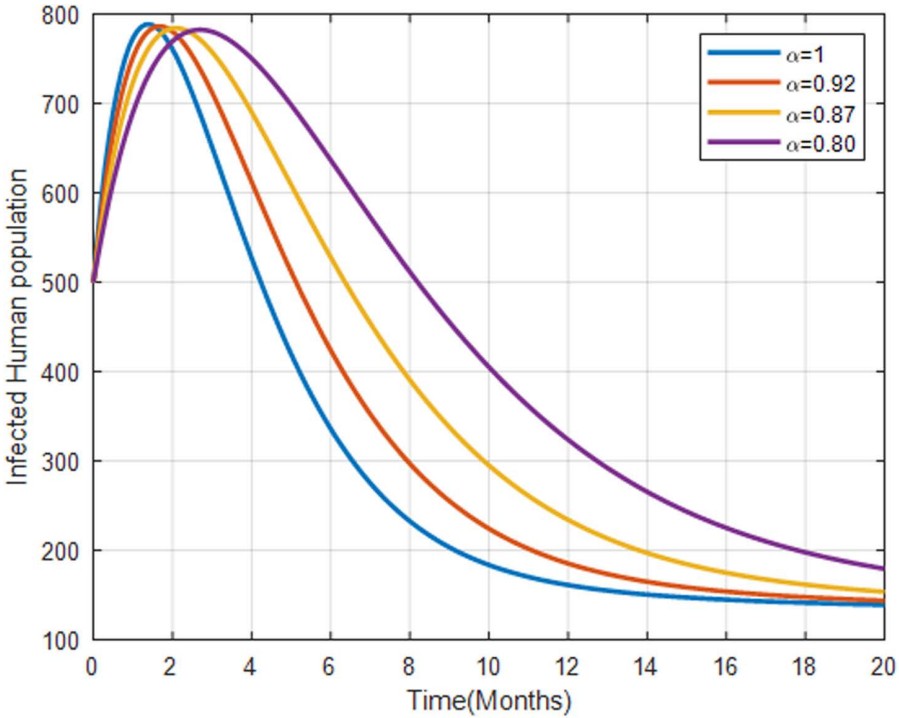

**Fig 12. Infected human population with fractional order $\alpha = 0.80$, $\alpha = 0.87$, $\alpha = 0.92$ and $\alpha = 1$.**

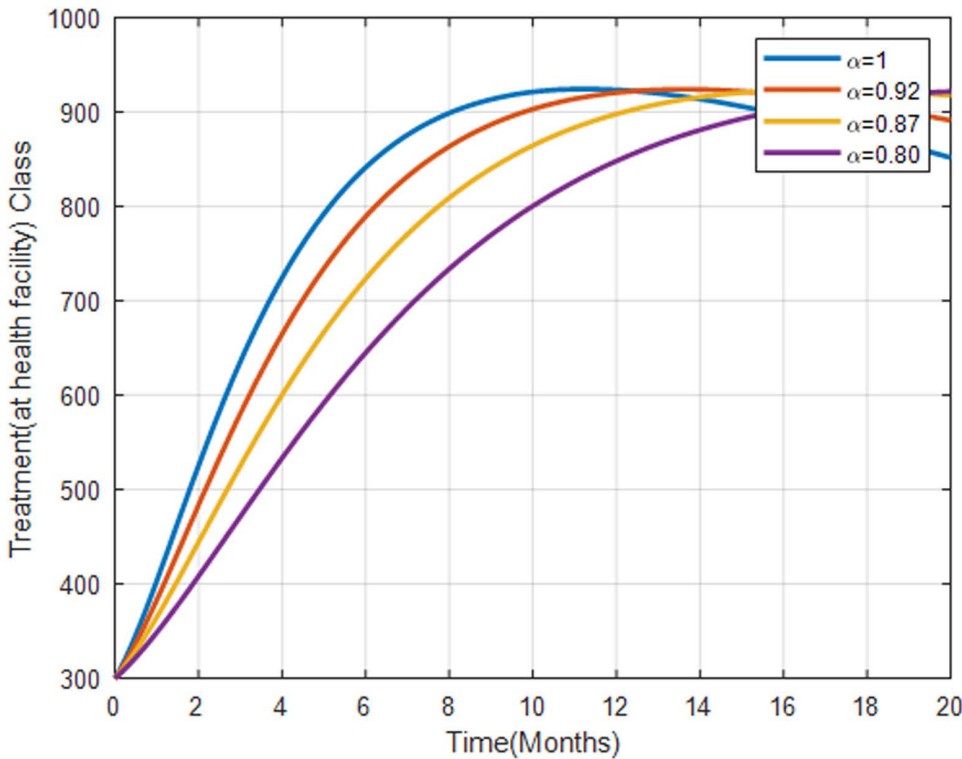

**Fig 13. Treatment class undergoing health facility with fractional order $\alpha = 0.8$, $\alpha = 0.87$, $\alpha = 0.92$, $\alpha = 1$.**

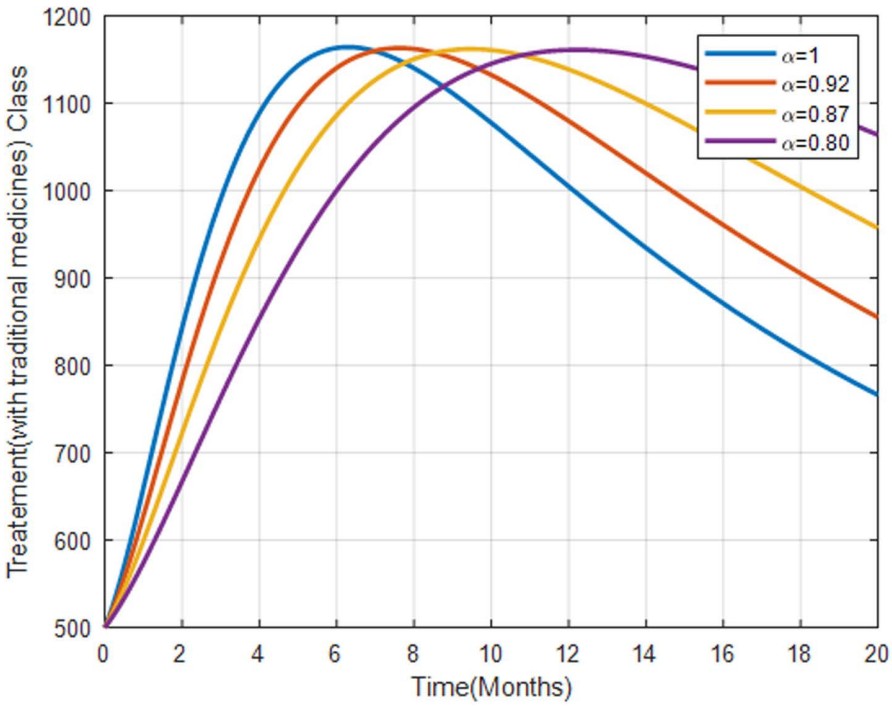

**Fig 14. Treatment class undergoing TMs with fractional order $\alpha = 0.8$, $\alpha = 0.87$, $\alpha = 0.92$, $\alpha = 1$.**

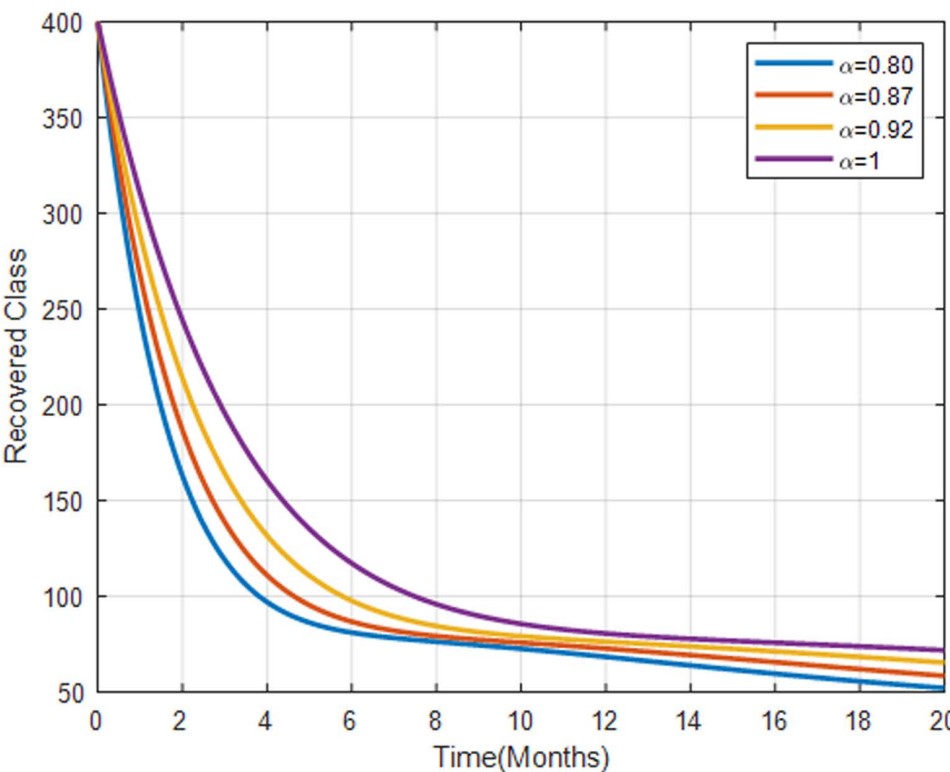

**Fig 15. Recovered human population with fractional order $\alpha = 0.80$, $\alpha = 0.87$, $\alpha = 0.92$ and $\alpha = 1$.**

traditional medicine and natural immunity. Moreover, a lower fractional order shows slower recovery, suggesting less effective treatment or higher relapse rates.

Fig 16 shows a decline in susceptible mosquitoes with increasing fractional orders, indicating effective transmission from mosquitoes to humans. This suggests successful transmission to humans, aligning with the observed increase in exposed and infected human populations.

Fig 17 shows that exposed mosquito population initially increases and then stabilizes with higher fractional orders. The higher fractional orders result in a quick increase in exposed mosquitoes, followed by stabilization, indicating a potential equilibrium in the transmission cycle.

Fig 18 shows a trend similar to that of exposed mosquitoes, increasing with higher fractional orders before stabilizing or decreasing. The infected mosquito population dynamics mirror those of the exposed mosquitoes, reflecting the transmission efficiency and the impact of control measures.

Moreover, in fractional-order dynamics, the order $\alpha$ significantly influences the behavior and memory effects of the system. Solutions in the range $[0.1,\ 0.8]$ may be impracticable due to memory and hereditary effects, insufficient sensitivity to initial conditions, numerical instability, biological irrelevance, and physical and interpretative constraints. For example, solutions in the range $[0.1,\ 0.8]$ may represent systems with unrealistically high memory effects, making them impractical for disease modeling. To better illustrate the impact, focus on $\alpha$ values that are biologically plausible and avoid computational pitfalls, ensuring a more realistic and interpretable representation of disease dynamics.

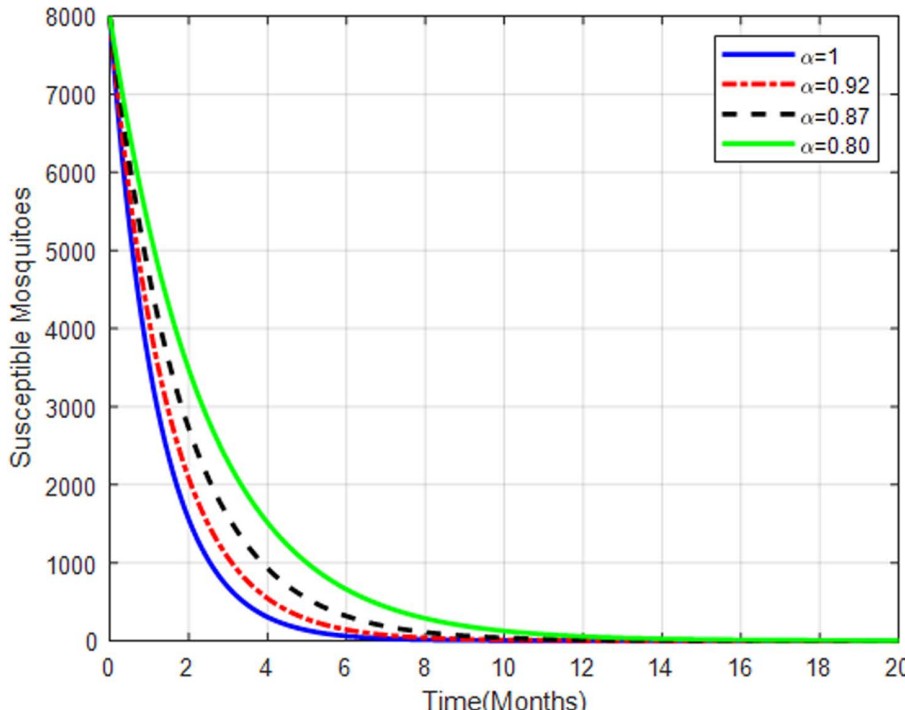

**Fig 16. Susceptible mosquitoes with fractional order $\alpha = 0.80,\ \alpha = 0.87,\ \alpha = 0.92$ and.$\alpha = 1.$**

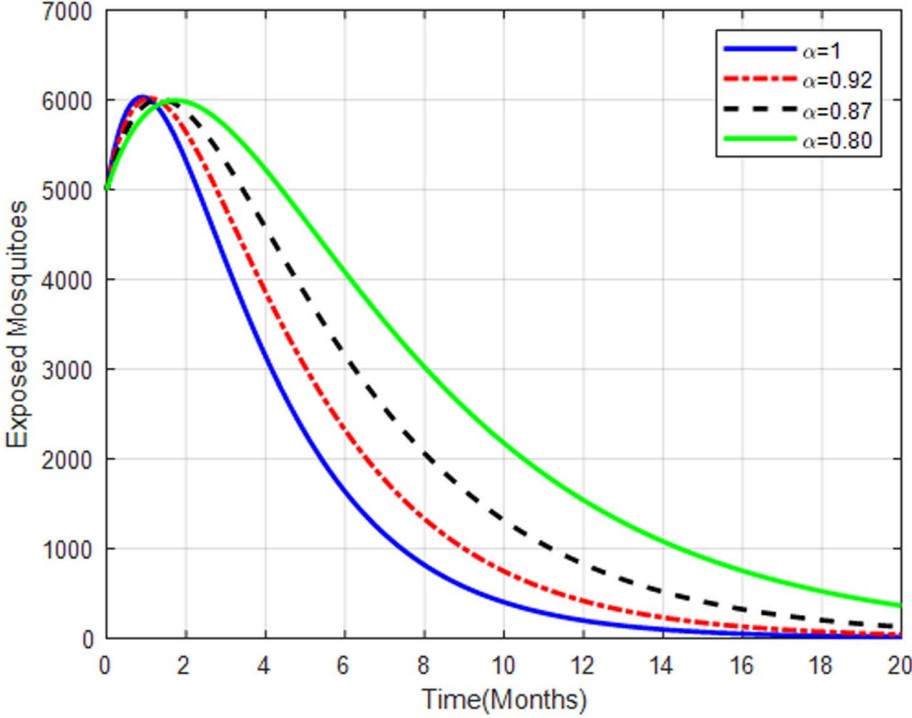

**Fig 17. Exposed mosquitoes with fractional order $\alpha = 0.80$, $\alpha = 0.87$, $\alpha = 0.92$ and $\alpha = 1$.**

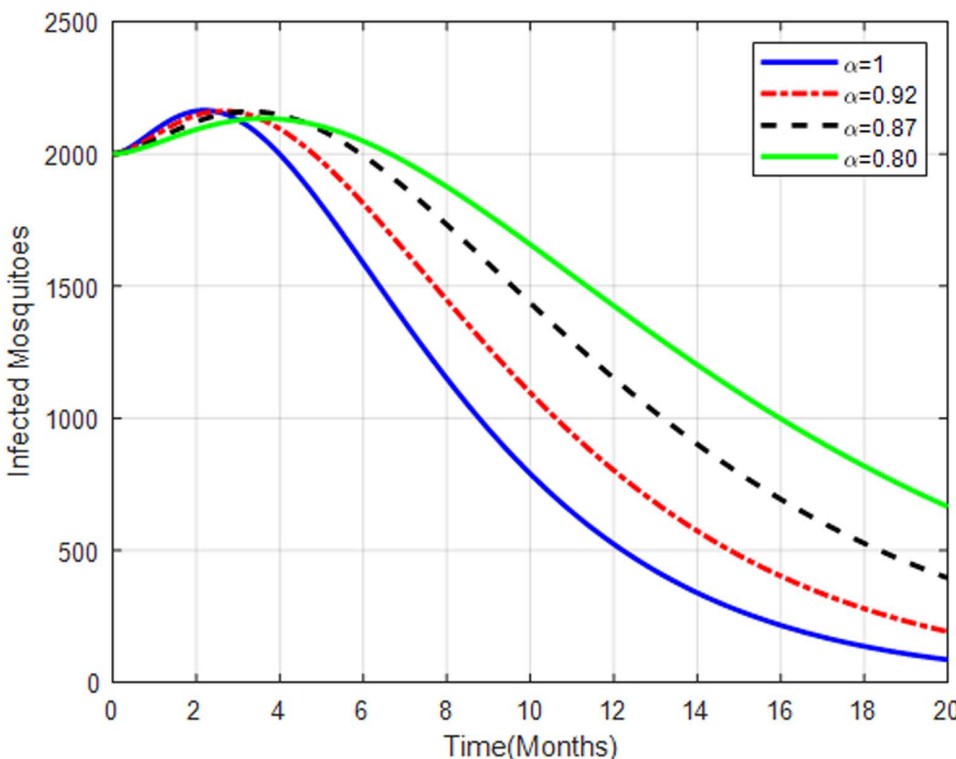

**Fig 18. Infected mosquitoes with fractional order $\alpha = 0.80$, $\alpha = 0.87$, $\alpha = 0.92$ and.$\alpha = 1$.**

## 7. Discussion

This study adds to the growing body of literature on the use of fractional calculus in epidemiological modeling, with a specific focus on malaria transmission. The application of the Atangana-Baleanu Caputo (ABC) fractional order in the model enables a more nuanced understanding of the disease's progression, particularly in settings where traditional medicine is prevalent. By incorporating the ABC fractional order, the model effectively captures the memory effects associated with treatment-seeking behaviors, highlighting the significance of past experiences in predicting future disease dynamics.

However, the model also brings to light challenges linked to the widespread use of traditional medicine, such as the lack of standardization and the potential for ineffective treatment. These issues can lead to prolonged infection periods and increased transmission rates, especially in vulnerable populations. The fractional-order model enhances the realism of malaria transmission simulations by accounting for the non-locality and memory effects characteristic of biological systems. This inclusion results in more accurate predictions of disease spread and provides insights into the potential impact of various intervention strategies.

The study's findings emphasize that, while traditional medicine plays an essential role in malaria treatment, collaboration with professional healthcare practices is crucial to ensure safe and effective outcomes. Integrating traditional and modern medical approaches, alongside rigorous scientific validation, can help optimize treatment protocols, reduce transmission rates, and improve overall public health outcomes in regions affected by malaria. Moreover, the findings from this study offer critical insights into shaping public health strategies, particularly in regions where traditional medicine is a prominent aspect of healthcare practices. The fractional-order model demonstrates that while traditional medicine plays a role in addressing malaria, its effectiveness can be undermined by a lack of standardization and clinical validation. This poses risks of incomplete treatment and potential relapse, which in turn may exacerbate malaria transmission. Public health strategies can leverage the model's findings by fostering collaboration between traditional medicine practitioners and professional healthcare providers.

## 8. Conclusion

In this paper, we present a fractional-order mathematical model based on the Atangana-Baleanu Caputo derivative to study malaria transmission dynamics, emphasizing the impact of treatment-seeking behavior in both professional healthcare and traditional medicine settings. We investigated the existence and uniqueness of solutions for the fractional-order model using the Banach fixed point theorem. Additionally, we explored the positivity and boundedness of the solutions to ensure the model's practicality and reliability. The model's equilibrium points were identified, revealing that the malaria-free and malaria-endemic equilibrium points are locally asymptotically stable for $\mathcal{R}_0 < 1$ and $\mathcal{R}_0 > 1$, respectively. The sensitivity analysis of the basic reproduction number suggests that the most critical factors in controlling malaria include reducing the mosquito biting rate, decreasing the mosquito recruitment rate, increasing the mosquito mortality rate, and enhancing treatment at professional health facilities.

Based on our findings, we recommend the following actions:

### i. Integrate traditional and modern medicine approaches

- **Policy recommendation**: Develop a national or regional framework for the integration of traditional medicine and modern healthcare practices, particularly in malaria-endemic regions. This could include training traditional practitioners on the importance of timely and effective treatment and promoting collaboration with professional healthcare workers.

- **Justification**: The study highlights the significance of past treatment-seeking behaviors and the memory effects of traditional medicine in predicting future malaria dynamics. By combining the strengths of both approaches, treatment protocols can be optimized, reducing delays in care and minimizing the spread of the disease.

### ii. Strengthen professional healthcare access and infrastructure

- **Policy recommendation**: Increase investments in health facility infrastructure, particularly in rural or remote areas where traditional medicine is more common. This could involve enhancing training for healthcare workers and ensuring that health facilities are equipped with effective anti-malarial treatments.

- **Justification**: The model emphasizes the importance of effective treatment at health facilities in controlling malaria. Inadequate access to professional healthcare may lead to prolonged infections and higher transmission rates, especially when ineffective traditional treatments are relied upon.

### iii. Public awareness campaigns on effective malaria treatment

- **Policy recommendation**: Implement targeted public health campaigns that educate communities about the risks of using clinically invalidated traditional remedies and the importance of seeking timely treatment from professional healthcare providers.

- **Justification**: The widespread use of non-standardized traditional medicine can contribute to prolonged infection periods and drug resistance. Educating the public on the risks associated with unproven treatments and promoting the use of evidence-based practices can reduce unnecessary delays in effective treatment.

### iv. Strengthen vector control measures

- **Policy recommendation**: Enhance malaria vector control programs by focusing on reducing the mosquito bite rate, increasing mosquito mortality, and decreasing mosquito recruitment rates. This could include the distribution of insecticide-treated bed nets, indoor spraying, and environmental management practices to eliminate mosquito breeding sites.

- **Justification**: The study's sensitivity analysis indicates that vector control is a critical factor in controlling malaria transmission. Effective mosquito control measures will significantly reduce the spread of the disease, complementing efforts to improve treatment outcomes.

### v. Improve data collection and monitoring systems

- **Policy recommendation**: Develop robust data collection and monitoring systems to track malaria incidence and treatment-seeking behaviors. This information can inform policy decisions and resource allocation.

- **Justification**: The model incorporates treatment-seeking behaviors, highlighting the need for accurate data to understand disease dynamics. Monitoring these behaviors and treatment outcomes will help policymakers design targeted interventions and track the impact of health strategies.

These recommendations aim to optimize malaria treatment, improve public health outcomes, and reduce transmission rates by leveraging both modern healthcare and traditional practices while addressing the risks posed by ineffective treatment methods.

## 9. Limitation of the research

This study provides valuable insights into malaria transmission dynamics, but it is essential to acknowledge limitations. These include simplified assumptions and a lack of empirical validation, which may not fully capture real-world malaria transmission dynamics. The model does not account for external factors like climate variability, human mobility, and vector behavior dynamics. Addressing these limitations in future studies could enhance the model's robustness and applicability, making it a more effective tool for malaria control and elimination strategies.

## Appendix 1. Proof of Routh-Hurwitz stability criterion.

$P_9 = 1, \; P_8 = (a_1 + b_2 + c_3 + d_4 + e_5 + f_6 + g_7 + h_8 + k_9)$

$$
\begin{aligned}
P_7 = (&a_1 b_2 + a_1 c_3 + b_2 c_3 + a_1 d_4 + b_2 d_4 + c_3 d_4 + a_1 e_5 + b_2 e_5 + c_3 e_5 + d_4 e_5 + a_1 f_6 + b_2 f_6 + c_3 f_6 + d_4 f_6 \\
&+ e_5 f_6 + a_1 g_7 + b_2 g_7 + c_3 g_7 + d_4 g_7 + e_5 g_7 + f_6 g_7 + a_1 h_8 + b_2 h_8 + c_3 h_8 + d_4 h_8 + e_5 h_8 + f_6 h_8 + g_7 h_8 + a_1 k_9 \\
&+ b_2 k_9 + c_3 k_9 + d_4 k_9 + e_5 k_9 + f_6 k_9 + g_7 k_9 + h_8 k_9)
\end{aligned}
$$

$$
\begin{aligned}
P_6 = &\; c_3 d_4 e_5 - c_6 e_3 f_5 + c_3 d_4 f_6 + c_3 e_5 f_6 + d_4 e_5 f_6 + c_3 d_4 g_7 + c_3 e_5 g_7 + d_4 e_5 g_7 + c_3 f_6 g_7 + d_4 f_6 g_7 \\
&+ e_5 f_6 g_7 + c_3 d_4 h_8 + c_3 e_5 h_8 + d_4 e_5 h_8 + c_3 f_6 h_8 + d_4 f_6 h_8 + e_5 f_6 h_8 + c_3 g_7 h_8 + d_4 g_7 h_8 + e_5 g_7 h_8 + f_6 g_7 h_8 \\
&+ (e_5 f_6 + e_5 g_7 + f_6 g_7 + (e_5 + f_6 + g_7) h_8 + d_4 (e_5 + f_6 + g_7 + h_8) + c_3 (d_4 + e_5 + f_6 + g_7 + h_8)) k_9 \\
&+ b_2 (e_5 f_6 + e_5 g_7 + f_6 g_7 + e_5 h_8 + f_6 h_8 + g_7 h_8 + (e_5 + f_6 + g_7 + h_8) k_9 + d_4 (e_5 + f_6 + g_7 \\
&+ h_8 + k_9) + c_3 (d_4 + e_5 + f_6 + g_7 + h_8 + k_9)) + a_1 (d_4 e_5 + d_4 f_6 + e_5 f_6 + d_4 g_7 + e_5 g_7 + f_6 g_7 \\
&+ d_4 h_8 + e_5 h_8 + f_6 h_8 + g_7 h_8 + (d_4 + e_5 + f_6 + g_7 + h_8) k_9 + c_3 (d_4 + e_5 + f_6 + g_7 \\
&+ h_8 + k_9) + b_2 (c_3 + d_4 + e_5 + f_6 + g_7 + h_8 + k_9))
\end{aligned}
$$

$$
\begin{aligned}
P_5 = &- a_4 b_1 c_2 d_3 + b_2 c_3 d_4 e_5 - b_2 c_6 e_3 f_5 - c_6 d_4 e_3 f_5 + b_2 c_3 d_4 f_6 + b_2 c_3 e_5 f_6 + b_2 d_4 e_5 f_6 + c_3 d_4 e_5 f_6 + b_2 c_3 d_4 g_7 \\
&+ b_2 c_3 e_5 g_7 + b_2 d_4 e_5 g_7 + c_3 d_4 e_5 g_7 - c_6 e_3 f_5 g_7 + b_2 c_3 f_6 g_7 + b_2 d_4 f_6 g_7 + c_3 d_4 f_6 g_7 + b_2 e_5 f_6 g_7 + c_3 e_5 f_6 g_7 \\
&+ d_4 e_5 f_6 g_7 + b_2 c_3 d_4 h_8 + b_2 c_3 e_5 h_8 + b_2 d_4 e_5 h_8 + c_3 d_4 e_5 h_8 - c_6 e_3 f_5 h_8 + b_2 c_3 f_6 h_8 + b_2 d_4 f_6 h_8 + c_3 d_4 f_6 h_8 \\
&+ b_2 e_5 f_6 h_8 + c_3 e_5 f_6 h_8 + d_4 e_5 f_6 h_8 + b_2 c_3 g_7 h_8 + b_2 d_4 g_7 h_8 + c_3 d_4 g_7 h_8 + b_2 e_5 g_7 h_8 + c_3 e_5 g_7 h_8 + d_4 e_5 g_7 h_8 \\
&+ b_2 f_6 g_7 h_8 + c_3 f_6 g_7 h_8 + d_4 f_6 g_7 h_8 + e_5 f_6 g_7 h_8 - b_9 c_2 h_3 k_8 + b_2 c_3 d_4 k_9 + b_2 c_3 e_5 k_9 + b_2 d_4 e_5 k_9 \\
&+ c_3 d_4 e_5 k_9 - c_6 e_3 f_5 k_9 + b_2 c_3 f_6 k_9 + b_2 d_4 f_6 k_9 + c_3 d_4 f_6 k_9 + b_2 e_5 f_6 k_9 + c_3 e_5 f_6 k_9 + d_4 e_5 f_6 k_9 + b_2 c_3 g_7 k_9 \\
&+ b_2 d_4 g_7 k_9 + c_3 d_4 g_7 k_9 + b_2 e_5 g_7 k_9 + c_3 e_5 g_7 k_9 + d_4 e_5 g_7 k_9 + b_2 f_6 g_7 k_9 + c_3 f_6 g_7 k_9 + d_4 f_6 g_7 k_9 \\
&+ e_5 f_6 g_7 k_9 + b_2 c_3 h_8 k_9 + b_2 d_4 h_8 k_9 + c_3 d_4 h_8 k_9 + b_2 e_5 h_8 k_9 + c_3 e_5 h_8 k_9 + d_4 e_5 h_8 k_9 + b_2 f_6 h_8 k_9 \\
&+ c_3 f_6 h_8 k_9 + d_4 f_6 h_8 k_9 + e_5 f_6 h_8 k_9 + b_2 g_7 h_8 k_9 + c_3 g_7 h_8 k_9 + d_4 g_7 h_8 k_9 + e_5 g_7 h_8 k_9 + f_6 g_7 h_8 k_9 \\
&+ a_1 (- c_6 e_3 f_5 + d_4 e_5 f_6 + d_4 e_5 g_7 + d_4 f_6 g_7 + e_5 f_6 g_7 + d_4 e_5 h_8 + d_4 f_6 h_8 + e_5 f_6 h_8 \\
&+ d_4 g_7 h_8 + e_5 g_7 h_8 + f_6 g_7 h_8 + d_4 e_5 k_9 + d_4 f_6 k_9 + e_5 f_6 k_9 + d_4 g_7 k_9 + e_5 g_7 k_9 + f_6 g_7 k_9 + d_4 h_8 k_9 \\
&+ e_5 h_8 k_9 + f_6 h_8 k_9 + g_7 h_8 k_9 + c_3 (f_6 g_7 + f_6 h_8 + g_7 h_8 + f_6 k_9 + g_7 k_9 + h_8 k_9 + e_5 (f_6 + g_7 + h_8 + k_9) \\
&+ d_4 (e_5 + f_6 + g_7 + h_8 + k_9)) + b_2 (e_5 f_6 + e_5 g_7 + f_6 g_7 + e_5 h_8 + f_6 h_8 + g_7 h_8 + e_5 k_9 + f_6 k_9 \\
&+ g_7 k_9 + h_8 k_9 + d_4 (e_5 + f_6 + g_7 + h_8 + k_9) + c_3 (d_4 + e_5 + f_6 + g_7 + h_8 + k_9)))
\end{aligned}
$$

$$
\begin{aligned}
P_4 = &- b_2 c_6 d_4 e_3 f_5 + b_2 c_3 d_4 e_5 f_6 + b_2 c_3 d_4 e_5 g_7 - b_2 c_6 e_3 f_5 g_7 - c_6 d_4 e_3 f_5 g_7 + b_2 c_3 d_4 f_6 g_7 + b_2 c_3 e_5 f_6 g_7 \\
&+ b_2 d_4 e_5 f_6 g_7 + c_3 d_4 e_5 f_6 g_7 + b_2 c_3 d_4 e_5 h_8 - b_2 c_6 e_3 f_5 h_8 - c_6 d_4 e_3 f_5 h_8 + b_2 c_3 d_4 f_6 h_8 + b_2 c_3 e_5 f_6 h_8 \\
&+ b_2 d_4 e_5 f_6 h_8 + c_3 d_4 e_5 f_6 h_8 + b_2 c_3 d_4 g_7 h_8 + b_2 c_3 e_5 g_7 h_8 + b_2 d_4 e_5 g_7 h_8 + c_3 d_4 e_5 g_7 h_8 - c_6 e_3 f_5 g_7 h_8 \\
&+ b_2 c_3 f_6 g_7 h_8 + b_2 d_4 f_6 g_7 h_8 + c_3 d_4 f_6 g_7 h_8 + b_2 e_5 f_6 g_7 h_8 + c_3 e_5 f_6 g_7 h_8 + d_4 e_5 f_6 g_7 h_8 \\
&- b_9 c_2 d_4 h_3 k_8 - b_9 c_2 e_5 h_3 k_8 - b_9 c_2 f_6 h_3 k_8 - b_9 c_2 g_7 h_3 k_8 - b_9 c_2 e_3 h_5 k_8 + b_9 c_2 g_3 h_7 k_8 \\
&+ b_1 c_2 (-a_6 e_3 f_5 + a_9 h_3 k_8) + (c_3 d_4 e_5 f_6 + c_3 d_4 e_5 g_7 + c_3 d_4 f_6 g_7 + c_3 e_5 f_6 g_7 + d_4 e_5 f_6 g_7 \\
&+ (e_5 f_6 g_7 + d_4 (e_5 f_6 + (e_5 + f_6) g_7) + c_3 (e_5 f_6 + (e_5 + f_6) g_7 + d_4 (e_5 + f_6 + g_7))) \\
&h_8 - c_6 e_3 f_5 (d_4 + g_7 + h_8) + b_2 (-c_6 e_3 f_5 + e_5 f_6 g_7 + (f_6 g_7 + e_5 (f_6 + g_7)) h_8 \\
&+ d_4 (g_7 h_8 + f_6 (g_7 + h_8) + e_5 (f_6 + g_7 + h_8)) + c_3 (f_6 g_7 (f_6 + g_7) h_8 + e_5 (f_6 + g_7 + h_8) \\
&+ d_4 (e_5 + f_6 + g_7 + h_8)))) k_9 - a_4 b_1 c_2 (d_5 e_3 + d_3 (e_5 + f_6 + g_7 + h_8 + k_9)) \\
&+ a_1 (c_3 d_4 e_5 f_6 + c_3 d_4 e_5 g_7 + c_3 d_4 f_6 g_7 + c_3 e_5 f_6 g_7 + d_4 e_5 f_6 g_7 + c_3 d_4 e_5 h_8 + c_3 d_4 f_6 h_8 \\
&+ c_3 e_5 f_6 h_8 + d_4 e_5 f_6 h_8 + c_3 d_4 g_7 h_8 + c_3 e_5 g_7 h_8 + d_4 e_5 g_7 h_8 + c_3 f_6 g_7 h_8 + d_4 f_6 g_7 h_8 \\
&+ e_5 f_6 g_7 h_8 - b_9 c_2 h_3 k_8 + (e_5 f_6 g_7 + (f_6 g_7 + e_5 (f_6 + g_7)) h_8 + d_4 (g_7 h_8 \\
&+ f_6 (g_7 + h_8) + e_5 (f_6 + g_7 + h_8)) + c_3 (f_6 g_7 + (f_6 + g_7) h_8 + e_5 (f_6 + g_7 + h_8) \\
&+ d_4 (e_5 + f_6 + g_7 + h_8))) k_9 - c_6 e_3 f_5 (d_4 + g_7 + h_8 + k_9) + b_2 (-c_6 e_3 f_5 + e_5 f_6 g_7 \\
&+ e_5 f_6 h_8 + e_5 g_7 h_8 + f_6 g_7 h_8 + (g_7 h_8 + f_6 (g_7 + h_8) + e_5 (f_6 + g_7 + h_8)) k_9 \\
&+ d_4 (g_7 h_8 + (g_7 + h_8) k_9 + f_6 (g_7 + h_8 + k_9) + e_5 (f_6 + g_7 + h_8 + k_9)) + c_3 (f_6 g_7 \\
&+ f_6 h_8 + g_7 h_8 + (f_6 + g_7 + h_8) k_9 + e_5 (f_6 + g_7 + h_8 + k_9) + d_4 (e_5 + f_6 + g_7 + h_8 + k_9))))
\end{aligned}
$$

$$P_3 = -a_4b_1c_2d_5e_3f_6 - a_4b_1c_2d_3e_5f_6 - a_4b_1c_2d_5e_3g_7 - a_4b_1c_2d_3e_5g_7 - b_2c_6d_4e_3f_5g_7 - a_4b_1c_2d_3f_6g_7$
$$+ b_2c_3d_4e_5f_6g_7 - a_4b_1c_2d_5e_3h_8 - a_4b_1c_2d_3e_5h_8 - b_2c_6d_4e_3f_5h_8 - a_4b_1c_2d_3f_6h_8 + b_2c_3d_4e_5f_6h_8 - a_4b_1c_2d_3g_7h_8$
$$+ b_2c_3d_4e_5g_7h_8 - b_2c_6d_4e_3f_5g_7h_8 - c_6d_4e_3f_5g_7h_8 + b_2c_3d_4f_6g_7h_8 + b_2c_3e_5f_6g_7h_8 + b_2d_4e_5f_6g_7h_8$
$$+ c_3d_4e_5f_6g_7h_8 + a_9b_1c_2d_4h_3k_8 + a_9b_1c_2e_5h_3k_8 - b_9c_2d_4e_5h_3k_8 + a_9b_1c_2f_6h_3k_8 - b_9c_2d_4f_6h_3k_8$
$$- b_9c_2e_5f_6h_3k_8 + a_9b_1c_2g_7h_3k_8 - b_9c_2d_4g_7h_3k_8 - b_9c_2e_5g_7h_3k_8 - b_9c_2f_6g_7h_3k_8 + a_9b_1c_2e_3h_5k_8$
$$- b_9c_2d_4e_3h_5k_8 - b_9c_2e_3f_6h_5k_8 - b_9c_2e_3g_7h_5k_8 - a_9b_1c_2g_3h_7k_8 + b_9c_2d_4g_3h_7k_8 + b_9c_2e_5g_3h_7k_8$
$$+ b_9c_2f_6g_3h_7k_8 + b_9c_2e_3g_5h_7k_8 + (d_4(-c_6e_3f_5 + c_3e_5f_6)g_7 + (d_4e_5f_6g_7 - c_6e_3f_5(d_4 + g_7)$
$$+ c_3(d_4e_5f_6 + (d_4e_5 + (d_4 + e_5)f_6)g_7))h_8 - a_4b_1c_2(d_5e_3 + d_3(e_5 + f_6 + g_7 + h_8)) + b_2(d_4e_5f_6g_7$
$$+ (e_5f_6g_7 + d_4(e_5f_6 + (e_5 + f_6)g_7))h_8 - c_6e_3f_5(d_4 + g_7 + h_8) + c_3(e_5f_6g_7 + (f_6g_7 + e_5(f_6 + g_7))h_8$
$$+ d_4(f_6g_7 + (f_6 + g_7)h_8 + e_5(f_6 + g_7 + h_8)))))k_9 - a_6b_1c_2e_3f_5(d_4 + g_7 + h_8 + k_9)$
$$+ -a_1(c_6d_4e_3f_5g_7 - c_3d_4e_5f_6g_7 + c_6d_4e_3f_5h_8 - c_3d_4e_5f_6h_8 - c_3d_4e_5g_7h_8 + c_6e_3f_5g_7h_8 - c_3d_4f_6g_7h_8 - c_3e_5f_6g_7h_8$
$$- d_4e_5f_6g_7h_8 + b_9c_2d_4h_3k_8 + b_9c_2e_5h_3k_8 + b_9c_2f_6h_3k_8 + b_9c_2g_7h_3k_8 + b_9c_2e_3h_5k_8 - b_9c_2g_3h_7k_8$
$$+ (-d_4e_5f_6g_7 - (e_5f_6g_7 + d_4(e_5f_6 + (e_5 + f_6)g_7))h_8 + c_6e_3f_5(d_4 + g_7 + h_8) + c_3(-d_4e_5f_6$
$$- (e_5f_6 + d_4(e_5 + f_6))g_7 - (e_5f_6 + (e_5 + f_6)g_7 + d_4(e_5 + f_6 + g_7))h_8))k_9$
$$+ b_2(-d_4e_5f_6g_7 - d_4e_5f_6h_8 - d_4e_5g_7h_8 - d_4f_6g_7h_8 - e_5f_6g_7h_8 - (f_6g_7h_8 + e_5(f_6g_7 + (f_6 + g_7)h_8)$
$$+ d_4(f_6g_7 + (f_6 + g_7)h_8 + e_5(f_6 + g_7 + h_8)))k_9 + c_6e_3f_5(d_4 + g_7 + h_8 + k_9)$
$$+ c_3(-d_4e_5f_6 - (e_5f_6 + d_4(e_5 + f_6))g_7 - (f_6g_7 + e_5(f_6 + g_7) + d_4(e_5 + f_6 + g_7))h_8$
$$- (f_6g_7 + (f_6 + g_7)h_8 + e_5(f_6 + g_7 + h_8) + d_4(e_5 + f_6 + g_7 + h_8))k_9)))$$

$$P_2 = -a_4b_1c_2d_5e_3f_6g_7 - a_4b_1c_2d_3e_5f_6g_7 - a_4b_1c_2d_5e_3f_6h_8 - a_4b_1c_2d_3e_5f_6h_8 - a_4b_1c_2d_5e_3g_7h_8$
$$- a_4b_1c_2d_3e_5g_7h_8 - b_2c_6d_4e_3f_5g_7h_8 - a_4b_1c_2d_3f_6g_7h_8 + b_2c_3d_4e_5f_6g_7h_8 + a_9b_1c_2d_4e_5h_3k_8$
$$+ a_9b_1c_2d_4f_6h_3k_8 + a_9b_1c_2e_5f_6h_3k_8 - b_9c_2d_4e_5f_6h_3k_8 + a_9b_1c_2d_4g_7h_3k_8 + a_9b_1c_2e_5g_7h_3k_8 - b_9c_2d_4e_5g_7h_3k_8 + a_9b_1c_2\ldots$
$$- b_9c_2d_4f_6g_7h_3k_8 - b_9c_2e_5f_6g_7h_3k_8 + a_9b_1c_2d_4e_3h_5k_8 + a_9b_1c_2e_3f_6h_5k_8 - b_9c_2d_4e_3f_6h_5k_8$
$$+ a_9b_1c_2e_3g_7h_5k_8 - b_9c_2d_4e_3g_7h_5k_8 - b_9c_2e_3f_6g_7h_5k_8 - a_9b_1c_2d_4g_3h_7k_8 - a_9b_1c_2e_5g_3h_7k_8$
$$+ b_9c_2d_4e_5g_3h_7k_8 - a_9b_1c_2f_6g_3h_7k_8 + b_9c_2d_4f_6g_3h_7k_8 + b_9c_2e_5f_6g_3h_7k_8 - a_9b_1c_2e_3g_5h_7k_8$
$$+ b_9c_2d_4e_3g_5h_7k_8 + b_9c_2e_3f_6g_5h_7k_8 - (b_2d_4(c_6e_3f_5 - c_3e_5f_6)g_7 + (d_4(c_6e_3f_5 - c_3e_5f_6)g_7$
$$+ b_2(-c_3d_4e_5f_6 - (c_3d_4e_5 + (c_3d_4 + (c_3 + d_4)e_5)f_6)g_7 + c_6e_3f_5(d_4 + g_7)))h_8$
$$+ a_4b_1c_2(d_5e_3(f_6 + g_7 + h_8) + d_3(f_6g_7 + (f_6 + g_7)h_8 + e_5(f_6 + g_7 + h_8))))k_9$
$$- a_6b_1c_2e_3f_5(h_8k_9 + g_7(h_8 + k_9) + d_4(g_7 + h_8 + k_9)) + a_1(c_3d_4e_5f_6g_7h_8 - b_9c_2d_4e_5h_3k_8 - b_9c_2d_4f_6h_3k_8$
$$- b_9c_2e_5f_6h_3k_8 - b_9c_2d_4g_7h_3k_8 - b_9c_2e_5g_7h_3k_8 - b_9c_2f_6g_7h_3k_8 - b_9c_2d_4e_3h_5k_8 - b_9c_2e_3f_6h_5k_8$
$$- b_9c_2e_3g_7h_5k_8 + b_9c_2d_4g_3h_7k_8 + b_9c_2e_5g_3h_7k_8 + b_9c_2f_6g_3h_7k_8 + b_9c_2e_3g_5h_7k_8$
$$+ (d_4e_5f_6g_7h_8 + c_3(e_5f_6g_7h_8 + d_4(e_5f_6g_7 + (e_5f_6 + (e_5 + f_6)g_7)h_8)))k_9 - c_6e_3f_5(g_7h_8k_9$
$$+ d_4(h_8k_9 + g_7(h_8 + k_9))) + b_2(d_4e_5f_6g_7h_8 + (e_5f_6g_7h_8 + d_4(e_5f_6g_7 + (e_5f_6 + (e_5 + f_6)g_7)h_8))k_9$
$$- c_6e_3f_5(g_7h_8 + (g_7 + h_8)k_9 + d_4(g_7 + h_8 + k_9)) + c_3(e_5f_6g_7h_8 + (f_6g_7h_8 + e_5(f_6g_7 + (f_6 + g_7)h_8))k_9$
$$+ d_4(f_6g_7h_8 + (g_7h_8 + f_6(g_7 + h_8))k_9 + e_5(g_7h_8 + (g_7 + h_8)k_9 + f_6(g_7 + h_8 + k_9))))))$$

$$P_1 = -a_4b_1c_2d_5e_3f_6g_7h_8 - a_4b_1c_2d_3e_5f_6g_7h_8 + a_9b_1c_2d_4e_5f_6h_3k_8 + a_9b_1c_2d_4e_5g_7h_3k_8 + a_9b_1c_2d_4f_6g_7h_3k_8$
$$+ a_9b_1c_2e_5f_6g_7h_3k_8 - b_9c_2d_4e_5f_6g_7h_3k_8 + a_9b_1c_2d_4e_3f_6h_5k_8 + a_9b_1c_2d_4e_3g_7h_5k_8 + a_9b_1c_2e_3f_6g_7h_5k_8$
$$- b_9c_2d_4e_3f_6g_7h_5k_8 - a_9b_1c_2d_4e_5g_3h_7k_8 - a_9b_1c_2d_4f_6g_3h_7k_8 - a_9b_1c_2e_5f_6g_3h_7k_8 + b_9c_2d_4e_5f_6g_3h_7k_8$
$$- a_9b_1c_2d_4e_3g_5h_7k_8 - a_9b_1c_2e_3f_6g_5h_7k_8 + b_9c_2d_4e_3f_6g_5h_7k_8 - (b_2d_4(c_6e_3f_5 - c_3e_5f_6)g_7h_8$
$$+ a_4b_1c_2(d_5e_3(f_6g_7 + (f_6 + g_7)h_8) + d_3(e_5f_6g_7 + (e_5f_6 + (e_5 + f_6)g_7)h_8)))k_9$
$$+ a_1(-b_9c_2(f_6(g_7(e_5h_3 + e_3h_5) - (e_5g_3 + e_3g_5)h_7) + d_4(e_5((f_6 + g_7)h_3 - g_3h_7)$
$$+ f_6(g_7h_3 + e_3h_5 - g_3h_7) + e_3(g_7h_5 - g_5h_7)))k_8 + d_4(-c_6e_3f_5 + c_3e_5f_6)g_7h_8k_9$
$$+ b_2(d_4(-c_6e_3f_5 + c_3e_5f_6)g_7h_8 + (d_4e_5f_6g_7h_8 + c_3(d_4e_5f_6g_7$
$$+ (d_4e_5f_6 + (d_4e_5 + (d_4 + e_5)f_6)g_7)h_8) - c_6e_3f_5(g_7h_8 + d_4(g_7 + h_8)))k_9))$
$$- a_6b_1c_2e_3f_5(g_7h_8k_9 + d_4(h_8k_9 + g_7(h_8 + k_9)))$$

$P_0 = a_1 b_9 c_2 d_4 f_6 k_8 e_5 g_3(\eta) + a_1 b_9 c_2 d_4 f_6 k_8 e_3 g_5(\eta) + a_9 b_1 c_2 d_4 f_6 k_8 e_5 g_3(\eta) a_9 b_1 c_2 d_4 f_6 k_8$
$e_3 g_5(\eta) + a_1 b_2 d_4 g_7 h_8 k_9 (c_6 e_3 f_5 - c_3 e_5 f_6) + a_4 b_1 c_2 f_6 g_7 h_8 k_9 (d_5 e_3 + d_3 e_5) + a_6 b_1 c_2 d_4 e_3 f_5 g_7 h_8 k_9 > 0$ Where, $c_6 e_3 f_5 > c_3 e_5 f_6$ i.e. $\gamma \gamma_1 \tau_2 > CEF$.

Since all the parameters in our model are positive and for $\mathcal{R}_0 > 1$ the positive endemic equilibrium point exists, then we can see that all the coefficients $P_i$ of the characteristic equation (21) are positive. To determine the sign of eigenvalues we use Routh – Hurwitz stability criteria. Now consider the following Routh - Hurwitz array:

$$
\begin{array}{l|ccccccccc}
\lambda^9 & P_9 & P_7 & P_5 & P_3 & P_1 & 0 & 0 & 0 & 0 \\
\lambda^8 & P_8 & P_6 & P_4 & P_2 & P_0 & 0 & 0 & 0 \\
\lambda^7 & b_1 & b_2 & b_3 & b_4 & 0 & 0 & 0 \\
\lambda^6 & c_1 & c_2 & c_3 & c_4 & 0 & 0 \\
\lambda^5 & d_1 & d_2 & d_3 & 0 & 0 \\
\lambda^4 & e_1 & e_2 & 0 & 0 \\
\lambda^3 & f_1 & f_2 & 0 \\
\lambda^2 & g_1 & 0 \\
\lambda^1 & h_1 \\
\lambda^0 & i_0
\end{array}
$$

Here, in the first column we have $P_9 > 0$, $P_8 > 0$ and we need to show: $b_1 = \frac{-1}{P_8} \begin{vmatrix} P_9 & P_7 \\ P_8 & P_6 \end{vmatrix} = \frac{P_7 P_8 - P_9 P_6}{P_8} > 0$,

for $P_7 P_8 > P_9 P_6$, $b_2 = \frac{-1}{P_8} \begin{vmatrix} P_9 & P_5 \\ P_8 & P_4 \end{vmatrix} = \frac{P_5 P_8 - P_9 P_4}{P_8} > 0$, for $P_5 P_8 > P_9 P_4$, $b_3 = \frac{-1}{P_6} \begin{vmatrix} P_7 & P_3 \\ P_6 & P_2 \end{vmatrix} = \frac{P_3 P_6 - P_7 P_2}{P_6} > 0$,

for $P_3 P_6 > P_7 P_2$, $b_4 = \frac{-1}{P_4} \begin{vmatrix} P_5 & P_1 \\ P_4 & P_0 \end{vmatrix} = \frac{P_4 P_1 - P_5 P_0}{P_4} > 0$, for $P_4 P_1 > P_5 P_0$, $c_1 = \frac{-1}{b_1} \begin{vmatrix} P_8 & P_6 \\ b_1 & b_2 \end{vmatrix} = \frac{b_1 P_6 - P_8 b_2}{b_1} > 0$, for

$b_1 P_6 > P_8 b_2$, $c_2 = \frac{-1}{b_1} \begin{vmatrix} P_8 & P_4 \\ b_1 & b_3 \end{vmatrix} = \frac{b_1 P_4 - P_8 b_3}{b_1} > 0$, for $b_1 P_4 > P_8 b_3$, $c_3 = \frac{-1}{b_2} \begin{vmatrix} P_6 & P_2 \\ b_2 & b_4 \end{vmatrix} = \frac{b_2 P_2 - P_6 b_4}{b_2} > 0$, for $b_2 P_2 > P_6 b_4$,

$c_3 = \frac{-1}{b_3} \begin{vmatrix} P_4 & P_0 \\ b_3 & 0 \end{vmatrix} = P_0 > 0$, $d_1 = \frac{-1}{c_1} \begin{vmatrix} b_1 & b_2 \\ c_1 & c_2 \end{vmatrix} = \frac{b_2 c_1 - b_1 c_2}{c_1} > 0$, for $b_2 c_1 > b_1 c_2$, $d_2 = \frac{-1}{c_1} \begin{vmatrix} b_1 & b_3 \\ c_1 & c_3 \end{vmatrix} = \frac{b_3 c_1 - b_1 c_3}{c_1} > 0$,

for $b_3 c_1 > b_1 c_3$, $d_3 = \frac{-1}{c_2} \begin{vmatrix} b_2 & b_4 \\ c_2 & c_4 \end{vmatrix} = \frac{b_4 c_2 - b_2 c_4}{c_2} > 0$, for $b_4 c_2 > b_2 c_4$, $e_1 = \frac{-1}{d_1} \begin{vmatrix} c_1 & c_2 \\ d_1 & d_2 \end{vmatrix} = \frac{d_1 c_2 - c_1 d_2}{d_1} > 0$, for $d_1 c_2 > c_1 d_2$,

$e_2 = \frac{-1}{d_1} \begin{vmatrix} c_1 & c_3 \\ d_1 & d_3 \end{vmatrix} = \frac{d_1 c_3 - c_1 d_3}{d_1} > 0$, for $d_1 c_3 > c_1 d_3$, $f_1 = \frac{-1}{e_1} \begin{vmatrix} d_1 & d_2 \\ e_1 & e_2 \end{vmatrix} = \frac{e_1 d_2 - d_1 e_2}{e_1} > 0$, for $e_1 d_2 > d_1 e_2$, $f_2 = \frac{-1}{e_1} \begin{vmatrix} d_1 & d_3 \\ e_1 & 0 \end{vmatrix} = d_3 > 0$,

$g_1 = \frac{-1}{f_1} \begin{vmatrix} e_1 & e_2 \\ f_1 & f_2 \end{vmatrix} = \frac{f_1 e_2 - e_1 f_2}{f_1} > 0$, for $f_1 e_2 > e_1 f_2$ and $h_1 = f_2 > 0$.

## Supporting information

**S1 Data. Parameters description of the model with their values.**
(DOCX)

## Acknowledgments

We would like to express our sincere gratitude to all those who contributed to the completion of this study. Special thanks to the Department of Mathematics of Debre Berhan University and Jimma University, Ethiopia for their support.

## Author contributions

**Conceptualization:** Sisay Fikadu Jaleta, Chernet Tuge Deressa.

**Data curation:** Sisay Fikadu Jaleta.

**Formal analysis:** Gemechis File Duressa, Chernet Tuge Deressa.

**Funding acquisition:** Sisay Fikadu Jaleta.

**Investigation:** Sisay Fikadu Jaleta, Chernet Tuge Deressa.

**Methodology:** Sisay Fikadu Jaleta, Gemechis File Duressa, Chernet Tuge Deressa.

**Project administration:** Gemechis File Duressa, Chernet Tuge Deressa.

**Resources:** Sisay Fikadu Jaleta.

**Software:** Sisay Fikadu Jaleta.

**Supervision:** Gemechis File Duressa, Chernet Tuge Deressa.

**Validation:** Gemechis File Duressa, Chernet Tuge Deressa.

**Visualization:** Sisay Fikadu Jaleta.

**Writing – original draft:** Sisay Fikadu Jaleta.

**Writing – review & editing:** Gemechis File Duressa, Chernet Tuge Deressa.

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
