## [Decision Letter · Decision Letter 0]

PONE-D-24-39540A Novel ABC Fractional-Order Mathematical Model for Malaria Transmission Dynamics Incorporating Treatment-Seeking Behavior

PLOS ONE

Dear Dr. Jaleta,

Thank you for submitting your manuscript to PLOS ONE. After careful consideration, we feel that it has merit but does not fully meet PLOS ONE’s publication criteria as it currently stands. Therefore, we invite you to submit a revised version of the manuscript that addresses the points raised during the review process.

We look forward to receiving your revised manuscript.

Kind regards,

Behzad Ghanbari

Academic Editor

PLOS ONE

*Comments from PLOS Editorial Office: We note that one or more reviewers has recommended that you cite specific previously published works. As always, we recommend that you please review and evaluate the requested works to determine whether they are relevant and should be cited. It is not a requirement to cite these works. We appreciate your attention to this request.*

Reviewers' comments:

Reviewer's Responses to Questions

**Comments to the Author**

1. Is the manuscript technically sound, and do the data support the conclusions?

Reviewer #1: Yes

Reviewer #2: Yes

Reviewer #3: Partly

2. Has the statistical analysis been performed appropriately and rigorously? 

Reviewer #1: Yes

Reviewer #2: Yes

Reviewer #3: N/A

3. Have the authors made all data underlying the findings in their manuscript fully available?

Reviewer #1: Yes

Reviewer #2: Yes

Reviewer #3: No

4. Is the manuscript presented in an intelligible fashion and written in standard English?

Reviewer #1: Yes

Reviewer #2: Yes

Reviewer #3: No

5. Review Comments to the Author

Reviewer #1: Review reports PONE-D-24-39540

The use of a fractional-order model is innovative, especially in the context of malaria transmission. Fractional-order models capture memory and hereditary properties, offering a deeper understanding of disease dynamics compared to traditional models. The inclusion of both professional health facilities and indigenous traditional medicine into the model is a significant strength. This dual approach adds realism, as these forms of treatment coexist in many malaria-endemic regions.

The authors provide a thorough mathematical exploration, including the existence and uniqueness of solutions, identification of equilibrium points (malaria-free and malaria-present), and detailed numerical simulations. This offers a solid theoretical foundation for the model. The study addresses a crucial public health issue, emphasizing the need for collaboration between traditional and professional healthcare practices, which is a relevant recommendation for policy-making in malaria-endemic regions. The analysis reveals that fractional-order effects play a crucial role in the progression of the disease. This finding is significant and opens up new avenues for understanding how the disease evolves under different treatment and intervention strategies.

Comments (minor/major) for improvement

1.While the study is mathematically rigorous, the narrative flow could benefit from better structure and clarity, particularly for readers unfamiliar with fractional calculus. A more accessible introduction to fractional-order models and their relevance in epidemiological studies might help.

2.What informs your choice of using the ABC operator in modeling the transmission of malaria? Highlight some of the useful applications of the ABC derivatives to modeling of real-life scenarios.

3.Reference materials on the ABC and other fractional operatorsis inadequate, addition of the following materials will boost the readership of your work: https://doi.org/10.1140/epjp/s13360-024-04910-z;
https://doi.org/10.32604/cmc.2020.011623
https://doi.org/10.1016/j.dajour.2022.100156;
https://doi.org/10.1016/j.chaos.2020.109826;
https://doi.org/10.1016/j.aej.2020.01.005;
https://doi.org/10.1016/j.physa.2019.123816;
https://doi.org/10.1016/j.chaos.2018.08.022;
https://doi.org/10.1016/j.chaos.2018.07.034;
https://doi.org/10.1016/j.chaos.2018.08.025

4.The numerical simulations section could be expanded with more detailed explanations of the algorithms or methods used. For example, specifying which numerical schemes were employed for solving the fractional-order system would enhance reproducibility.

5.While the model includes various parameters, a formal sensitivity analysis would be beneficial. Understanding which parameters (such as the rate of treatment or transmission rates) most significantly affect disease dynamics would provide valuable insights into which interventions are most effective.

6.The role of indigenous traditional medicine is mentioned, but the discussion on how it affects long-term disease management could be expanded. Are there any specific case studies or data that support the claims about temporary recovery and relapse? This would add depth to the discussion on traditional medicine's role in malaria treatment.

7.The study would be stronger if it included a comparison of the model’s predictions with actual data from malaria-endemic regions. Incorporating data-driven validation would further substantiate the conclusions drawn from the model.

8.The paper provides a valuable discussion on public health initiatives, but more concrete policy recommendations based on the model’s findings could be included. This would help in translating the mathematical results into actionable health strategies.

Minor concerns

•It would be useful to include a clearer or visualizations of the model's numerical simulations. There are some cases where tiny lines were used.

•It is obvious that solution below =0.8 appears invisible, explain what makes solution in the range [0.1, 0.8] impracticable. This would help to better illustrate how the fractional-order dynamics impact disease progression over time.

•Ensure that all variables and parameters are consistently defined, and consider simplifying the notation where possible to avoid overwhelming the reader.

Finally, this paper presents a novel and important contribution to the study of malaria transmission dynamics using the concept of fractional calculus. The model’s incorporation of treatment-seeking behavior and traditional medicine interventions reflects real-world complexities and offers valuable insights for public health strategies. With clearer exposition, sensitivity analysis, and validation with empirical data, the manuscript has the potential to make a significant impact in the field of epidemiological modeling. I would recommend a revision for this work.

Reviewer #2: The manuscript titled "A Novel ABC Fractional-Order Mathematical Model for Malaria Transmission Dynamics Incorporating Treatment-Seeking Behavior" presents an interesting and valuable contribution to the study of malaria dynamics using fractional-order calculus, particularly by integrating the important aspect of treatment-seeking behavior. The inclusion of both professional healthcare and indigenous traditional medicine is particularly noteworthy, as it adds a real-world complexity to the model, making it more applicable to developing regions where such dual treatment paradigms are prevalent. The mathematical analysis, focusing on equilibrium points and the effects of fractional orders, appears sound, and the authors' numerical simulations provide compelling insights into how fractional-order models can offer a more detailed understanding of disease progression compared to traditional integer-order models. However, there are several areas that require clarification and further development. Firstly, the authors should provide more detailed explanations of the biological significance of the fractional-order terms, as this would enhance the paper’s accessibility to a broader audience. Additionally, the paper would benefit from a more thorough sensitivity analysis, exploring how changes in key parameters impact the model’s outcomes. From a structural perspective, the manuscript could also be improved by refining the presentation of numerical results, ensuring a more cohesive discussion of the model's implications in the real world by comparing the numerical results with real data. The authors are also advised to consider citing relevant recent developments in fractional calculus, particularly the following works: A new fractional model and optimal control of a tumor-immune surveillance with non-singular derivative operator, Chaos: An Interdisciplinary Journal of Nonlinear Science, 29(8), 083127, 2019; Stability analysis and system properties of Nipah virus transmission: A fractional calculus case study, Chaos, Solitons & Fractals, 166, 112990, 2023; Fractional treatment: an accelerated mass-spring system, Romanian Reports in Physics, 74(4), 122, 2022; Dynamical behaviours and stability analysis of a generalized fractional model with a real case study, Journal of Advanced Research, 48, 157-173, 2023. These studies could provide additional context and support for the use of fractional calculus in modeling complex systems. Overall, while the paper has significant potential, I recommend major revisions to address these issues and strengthen the manuscript’s contribution to the field.

Reviewer #3: The article is well organized. The viewpoint in this article and the results have merit and are of interest, thus it will be reasonable to consider the publication of this paper.

I recommend acceptance of the paper for publication subject to the above minor changes aimed at improving the quality of the article.

6. PLOS authors have the option to publish the peer review history of their article (what does this mean? ). If published, this will include your full peer review and any attached files.

**Do you want your identity to be public for this peer review?** For information about this choice, including consent withdrawal, please see our Privacy Policy .

Reviewer #1: **Yes: ** Kolade M. Owolabi

Reviewer #2: No

Reviewer #3: No

---

## [Author Response · Author response to Decision Letter 1]

7 Dec 2024

Dear editor and Reviewers,

Thank you for the opportunity to revise and resubmit our manuscript, titled “A Novel ABC Fractional-Order Mathematical Model for Malaria Transmission Dynamics Incorporating Treatment-Seeking Behavior." We are grateful for the constructive feedback provided, which has greatly improved the quality and clarity of our work. Below, we provide a detailed point-by-point response to each comment, highlighting the changes made in the revised manuscript.

Reviewer#1 Comments and our responses:

1. While the study is mathematically rigorous, the narrative flow could benefit from better structure and clarity, particularly for readers unfamiliar with fractional calculus. A more accessible introduction to fractional-order models and their relevance in epidemiological studies might help.

Response: Thank you for your suggestion. We have revised the introduction to include a more accessible explanation of fractional calculus on page 2, section 1 (paragraph 2), and we have provided here the summary of our revision as follows:

Fractional calculus has revolutionized the modeling of complex systems and real-world phenomena, extending traditional calculus to deal with derivatives and integrals of non-integer orders. It captures memory dynamics and genetic properties in biological and engineering systems, expanding stability domains. In epidemiology, fractional-order models are gaining attention due to their ability to describe memory effects, allowing for a weighted understanding of the current state based on past conditions. This adaptability is particularly relevant for malaria, where factors like delayed immune responses, treatment adherence, and environmental influences can be better modeled using fractional derivatives.

2. What informs your choice of using the ABC operator in modeling the transmission of malaria? Highlight some of the useful applications of the ABC derivatives to modeling of real-life scenarios.

Response: We appreciate this feedback and have incorporated a detailed explanation of the ABC operator in Section 1 (paragraph 3) of the revised manuscript. The additional content highlights its suitability for modeling systems with memory and hereditary properties, its non-singular kernel, and its applicability to real-world scenarios such as viscoelastic materials, finance, thermodynamics, and disease modeling.

3. Reference materials on the ABC and other fractional operators is inadequate, addition of the following materials will boost the readership of your work.

Response: We have reviewed the suggested references and included relevant citations to strengthen our manuscript.

4. The numerical simulations section could be expanded with more detailed explanations of the algorithms or methods used. For example, specifying which numerical schemes were employed for solving the fractional-order system would enhance reproducibility.

Response: We have revised the Numerical Simulations section, on page 18 (Section 6), to provide a detailed explanation of the numerical schemes by Toufik and Atangana, incorporating the use of Lagrange polynomials and fractional calculus fundamentals. MATLAB R2023a was used for simulations, and details on initial conditions and parameters are included.

5. While the model includes various parameters, a formal sensitivity analysis would be beneficial. Understanding which parameters (such as the rate of treatment or transmission rates) most significantly affect disease dynamics would provide valuable insights into which interventions are most effective.

Response: We have added a detailed sensitivity analysis and presented the results in Section 5 (page 18). This analysis highlights the most sensitive parameters affecting the basic reproduction number R_0, emphasizing their implications for control measures.

6. The role of indigenous traditional medicine is mentioned, but the discussion on how it affects long-term disease management could be expanded. Are there any specific case studies or data that support the claims about temporary recovery and relapse? This would add depth to the discussion on traditional medicine's role in malaria treatment.

Response: We have expanded this discussion in section 1 (in paragraph 4), citing studies such as Makundi et al. on the impact of traditional healers on malaria outcomes and other data supporting the role of traditional medicine in relapse and temporary recovery. The revised manuscript includes a more detailed discussion of traditional medicine's role in malaria treatment, supported by case studies from Tanzania and references to studies on herbal remedies. This discussion highlights temporary recovery, relapse dynamics, and the importance of integrating traditional practices with modern healthcare.

7. The study would be stronger if it included a comparison of the model’s predictions with actual data from malaria-endemic regions. Incorporating data-driven validation would further substantiate the conclusions drawn from the model.

Response: Although we couldn't collect primary data due to the unavailability of organized data in Ethiopia, we have used relevant secondary data from credible literature sources, which have been properly cited and acknowledged in the manuscript. These secondary data have been used to validate our model. We believe this approach allows us to draw meaningful conclusions and contributes to the understanding of malaria trends in the region. Accordingly, we have added a new section (section 9) in our revised manuscript discussing the limitations of our study.

8. The paper provides a valuable discussion on public health initiatives, but more concrete policy recommendations based on the model’s findings could be included. This would help in translating the mathematical results into actionable health strategies.

Response: Based on our findings, we have included actionable policy recommendations in section 8 (page 29), focusing on integrating traditional and modern medicine, strengthening healthcare infrastructure, and promoting public awareness campaigns. These recommendations are derived from the sensitivity analysis and model insights.

Minor concerns

It would be useful to include a clearer or visualizations of the model's numerical simulations. There are some cases where tiny lines were used.

Response: Thank you for your feedback, and we have improved the figures for clarity in the revised manuscript.

It is obvious that solution below α=0.8 appears invisible, explain what makes solution in the range [0.1, 0.8] impracticable. This would help to better illustrate how the fractional-order dynamics impact disease progression over time.

Response: Thank you for your feedback, and a detailed discussion on the impracticality of solutions in the range [0.1,0.8] is included in section 7 (page 28), addressing memory effects, biological irrelevance, and numerical instability.

Ensure that all variables and parameters are consistently defined, and consider simplifying the notation where possible to avoid overwhelming the reader.

Response: Thank you for your feedback, and we ensured consistent definitions and streamlined notation to make the manuscript more reader-friendly.

We hope these revisions address all the concerns raised by the reviewer. We sincerely thank the reviewer and the editorial team for their valuable feedback, which has significantly strengthened our manuscript. The manuscript is now better aligned with the goals of PLOS ONE, providing insights into both the theoretical and practical aspects of malaria transmission dynamics.

Reviewer #2 Comments and our responses

The manuscript titled "A Novel ABC Fractional-Order Mathematical Model for Malaria Transmission Dynamics Incorporating Treatment-Seeking Behavior" presents an interesting and valuable contribution to the study of malaria dynamics using fractional-order calculus, particularly by integrating the important aspect of treatment-seeking behavior. The inclusion of both professional healthcare and indigenous traditional medicine is particularly noteworthy, as it adds a real-world complexity to the model, making it more applicable to developing regions where such dual treatment paradigms are prevalent. The mathematical analysis, focusing on equilibrium points and the effects of fractional orders, appears sound, and the authors' numerical simulations provide compelling insights into how fractional-order models can offer a more detailed understanding of disease progression compared to traditional integer-order models. However, there are several areas that require clarification and further development. Firstly, the authors should provide more detailed explanations of the biological significance of the fractional-order terms, as this would enhance the paper’s accessibility to a broader audience. Additionally, the paper would benefit from a more thorough sensitivity analysis, exploring how changes in key parameters impact the model’s outcomes. From a structural perspective, the manuscript could also be improved by refining the presentation of numerical results, ensuring a more cohesive discussion of the model's implications in the real world by comparing the numerical results with real data. The authors are also advised to consider citing relevant recent developments in fractional calculus, particularly the following works:

Response: We sincerely thank you for your encouraging comments about our work. We have carefully addressed all the concerns raised and revised the manuscript accordingly. The changes are detailed below:

Biological significance of fractional-order terms: In response to your suggestion, we have included a detailed explanation of the biological significance of fractional-order terms in section 1 (paragraph 2). This addition highlights how fractional-order terms reflect processes involving memory and history-dependent dynamics, which are particularly relevant to real-world phenomena such as disease progression and immune response. These revisions are included in the introduction ensuring accessibility to a broader audience.

Sensitivity analysis concern: We have extended the sensitivity analysis by examining the effects of variations in key parameters on the model's outcomes. Specifically, we have clarified in the revised manuscript (in section 5) how parameters in the model impact the basic reproduction number.

Numerical results and real-world implications (in section 6): Here, we would like to highlight that we have used MATLAB R2023a with the ODE45 method to obtain our numerical results, ensuring accurate simulations of malaria transmission dynamics. Additionally, we have included a dedicated section in the revised manuscript discussing recommendations for communities based on our findings. These recommendations aim to support evidence-based interventions and strategies for malaria control in the context of limited data availability. We believe these revisions enhance the real-world applicability of our study and its potential to inform public health strategies in affected regions. Recognizing the importance of transparency, we have included a section outlining the limitations of our study. These include challenges in parameter estimation, assumptions related to treatment-seeking behavior, and potential extensions of the model to incorporate more complex transmission dynamics. This addition sets the stage for future research while maintaining the integrity of the current work.

Citations of recent developments in fractional calculus: We have incorporated citations to recent advancements in fractional calculus, as suggested. These references strengthen the manuscript's foundation and situate our work within the broader context of emerging research in the field.

We hope these revisions address all the concerns raised by the reviewer. We sincerely thank the reviewer and the editorial team for their valuable feedback, which has significantly strengthened our manuscript. The manuscript is now better aligned with the goals of PLOS ONE, providing insights into both the theoretical and practical aspects of malaria transmission dynamics.

Reviewer #3 Comments and our responses

In this paper, the authors have introduced an advanced mathematical model to analyse malaria transmission. It focuses on integrating the Atangana-Baleanu Caputo (ABC) fractional-order derivative into a malaria model. The contents of the paper are good and contain new ideas. Anyhow, I would like to see the following modifications, whether minor or major, in the revised version, which would increase the strength of the paper and increase its potential readers, as well as improve the current work.

1. The abstract is dense and could benefit from more clarity, especially in explaining the novelty and significance of the approach. Simplifying the language and explicitly stating the problem, solution, and main contributions could make it more accessible.

Response: Thank you for your feedback, and we have rewritten the abstract to simplify the language and enhance clarity, explicitly highlighting the problem, solution, and main contributions. The revised abstract now focuses on the novelty and significance of incorporating the ABC fractional-order derivative in our malaria transmission model and its implications for public health strategies.

2. The introduction is comprehensive but might overwhelm the reader with too much background information. Consider condensing the literature review and focusing more on the specific gaps this paper addresses. The introduction should end with a clear statement of the research objectives and contributions.

Response: Thank you for your feedback, and the introduction has been condensed [in section 1 of the revised manuscript] to streamline the background information. We have emphasized the specific research gaps and concluded the section with a concise statement of our objectives and contributions, as per your suggestion.

3. The transitions between sections are abrupt.

Response: Thank you for this suggestion. We have carefully revised the manuscript to include logical linking sentences between sections, ensuring smoother transitions and better readability.

4. Include more detailed information about the software or programming languages used for simulations and ensure that all mathematical techniques can be replicated by other researchers.

Response: Thank you for this suggestion. We have added a detailed description of the numerical techniques and the software used for simulations in the revised manuscript section [Section 6]. Specifically, the revised manuscript mentions that MATLAB R2023a was utilized for numerical simulations and provides initial population values and parameter settings to facilitate replication by other researchers.

5. Add a section on limitations, such as potential biases in the data, challenges in modelling human behaviour, or the applicability of the model to other diseases.

Response: We appreciate your suggestion. We have added a section discussing the study's limitations in the revised manuscript of section [Section 9]. The limitations section addresses potential biases in secondary data, challenges in modeling human behavior, and the applicability of our model to other diseases.

6. Strengthen citation practices by referencing foundational work where applicable, particularly for sections that discuss mathematical techniques and the role of traditional medicine in malaria treatment.

Response: Thank you for your feedback. We have included additional citations to foundational work in both mathematical modeling techniques and the discussion of traditional medicine's role in malaria treatment.

7. Include a more thorough discussion of how the model’s findings can inform public health strategies, particularly for integrating traditional and modern medicine.

Response: Thank you for this suggestion. The discussion section has been expanded in the revised manuscript [Section 7, page 29] to elaborate on the implications of our findings for public health strategies. We emphasize the importance of integrating traditional and modern medicine to ensure effective and safe malaria treatment, highlighting potential policy recommendations and collaborative approaches.

We hope these revisions address all the concerns raised by the reviewer. We sincerely

---

## [Decision Letter · Decision Letter 1]

A Novel ABC Fractional-Order Mathematical Model for Malaria Transmission Dynamics Incorporating Treatment-Seeking Behavior

PONE-D-24-39540R1

Dear Dr. Jaleta,

We’re pleased to inform you that your manuscript has been judged scientifically suitable for publication and will be formally accepted for publication once it meets all outstanding technical requirements.

Kind regards,

Behzad Ghanbari

Academic Editor

PLOS ONE

Additional Editor Comments (optional):

Reviewers' comments:

Reviewer's Responses to Questions

**Comments to the Author**

1. If the authors have adequately addressed your comments raised in a previous round of review and you feel that this manuscript is now acceptable for publication, you may indicate that here to bypass the “Comments to the Author” section, enter your conflict of interest statement in the “Confidential to Editor” section, and submit your "Accept" recommendation.

Reviewer #1: All comments have been addressed

Reviewer #3: All comments have been addressed

2. Is the manuscript technically sound, and do the data support the conclusions?

Reviewer #1: Yes

Reviewer #3: Partly

3. Has the statistical analysis been performed appropriately and rigorously? 

Reviewer #1: Yes

Reviewer #3: Yes

4. Have the authors made all data underlying the findings in their manuscript fully available?

Reviewer #1: Yes

Reviewer #3: (No Response)

5. Is the manuscript presented in an intelligible fashion and written in standard English?

Reviewer #1: Yes

Reviewer #3: (No Response)

6. Review Comments to the Author

Reviewer #1: Review Report

The study presents a novel approach to modeling malaria transmission dynamics using an ABC fractional-order model that incorporates treatment-seeking behavior, including professional health interventions and traditional medicine practices. This work is timely and significant given the global burden of malaria, particularly in developing regions. By introducing fractional-order dynamics, the authors provide a more refined understanding of disease progression and treatment efficacy.

The manuscript is well-written, and the theoretical framework is sound, supported by appropriate mathematical analysis and numerical simulations. However, certain aspects can be clarified or expanded to enhance the overall quality and impact of the work.

In terms of contribution: The integration of fractional-order dynamics with treatment-seeking behavior is innovative and offers new insights into malaria transmission modeling.

In terms of mathematical challenge: The study thoroughly investigates the existence and uniqueness of solutions, and equilibrium points (malaria-free and malaria-present) are well-characterized.

Relevance to public health: By addressing the dichotomy between professional healthcare and traditional medicine, the study underscores a crucial issue in malaria management.

Policy and implications: The recommendations for fostering collaboration between traditional medicine practitioners and professional healthcare providers are actionable and relevant.

This manuscript offers a significant contribution to the field of mathematical epidemiology by introducing a fractional-order model to better understand malaria transmission dynamics and the role of treatment-seeking behavior. The theoretical results and policy recommendations are highly relevant. I recommend the paper for publication in PONE

Reviewer #3: This paper is fascinating and challenging. Moreover, this paper is well-organized and well-written. In my opinion, the paper deserves publication in the journal.

7. PLOS authors have the option to publish the peer review history of their article (what does this mean? ). If published, this will include your full peer review and any attached files.

**Do you want your identity to be public for this peer review?** For information about this choice, including consent withdrawal, please see our Privacy Policy .

Reviewer #1: **Yes: ** Kolade Matthew Owolabi

Reviewer #3: No

---

## [Editor Report · Acceptance letter]

PONE-D-24-39540R1

PLOS ONE

Dear Dr. Jaleta,

I'm pleased to inform you that your manuscript has been deemed suitable for publication in PLOS ONE. Congratulations! Your manuscript is now being handed over to our production team.

Kind regards,

on behalf of

Professor Behzad Ghanbari

Academic Editor

PLOS ONE